# Resolving Training Biases via Influence-Based Data Relabeling

**Shuming Kong, Yanyan Shen, Linpeng Huang**
Department of Computer Science and Engineering
Shanghai Jiao Tong University
{leinuo123,shenyy,lphuang}@sjtu.edu.cn

## Abstract

The performance of supervised learning methods easily suffers from the training bias issue caused by train-test distribution mismatch or label noise. Influence function is a technique that estimates the impacts of a training sample on the model's predictions. Recent studies on *data resampling* have employed influence functions to identify *harmful* training samples that will degrade model's test performance. They have shown that discarding or downweighting the identified harmful training samples is an effective way to resolve training biases. In this work, we move one step forward and propose an influence-based relabeling framework named RDIA for reusing harmful training samples toward better model performance. To achieve this, we use influence functions to estimate how relabeling a training sample would affect model's test performance and further develop a novel relabeling function $\mathcal{R}$. We theoretically prove that applying $\mathcal{R}$ to relabel harmful training samples allows the model to achieve lower test loss than simply discarding them for any classification tasks using cross-entropy loss. Extensive experiments on ten real-world datasets demonstrate RDIA outperforms the state-of-the-art data resampling methods and improves model's robustness against label noise.

## 1 Introduction

Training data plays an inevitably important role in delivering the model's final performance. It has been well recognized that the training bias issue will compromise model performance to a large extent (Arpit et al., 2017). Specifically, there are two major scenarios where training biases show up. The first and most common scenario is that training samples involve corrupted labels that could be originated at possibly every step of the data lifecycle (Anderson & McGrew, 2017; Dolatshah et al., 2018; Pei et al., 2020; Yu & Qin, 2020). The second scenario is that the training and test sets are sampled from the respective distributions $P_{train}(x, y)$ and $P_{test}(x, y)$, but $P_{train}$ is different from $P_{test}$ (Guo et al., 2020; He & Garcia, 2009; Zou et al., 2019). Both corrupted labels and distribution mismatch will hurt the generalization ability of a trained model (Fang et al., 2020; Zhang et al., 2017; Chen et al., 2021). We generally refer to training samples with corrupted labels or those inducing distribution mismatch as *harmful* samples.

Data resampling is a widely used strategy to deal with harmful training samples. Existing resampling approaches (Chawla et al., 2002; Mahmoody et al., 2016; Malach & Shalev-Shwartz, 2017; Ren et al., 2018) propose to assign different weights to training samples, which aim to mitigate the negative impacts of harmful samples on model's generalization ability. Among them, most resampling approaches (Arazo et al., 2019; Han et al., 2018; Li et al., 2020; Wang et al., 2020a) rely on training loss to identify harmful samples from the whole training set. They follow the insight that the samples with higher training losses are very likely to have corrupted labels, and hence it is often beneficial to downweight them during the process of model training. However, such loss-based resampling methods have two limitations. First, they are only able to deal with the training biases caused by training samples with corrupted labels (aka noisy samples). Second, the small-loss trick typically holds true for deep models but not for any predictive models (Zhang et al., 2017). To address the limitations, one recent work (Wang et al., 2020b) proposes a new resampling scheme based on influence functions (Cook & Weisberg, 1980). The idea is to estimate the influence of each training sample on model's predictions over the test set. Any training samples that would cause an

increase in the test loss are considered as harmful and will be downweighted afterwards. It is worth mentioning that the influence functions have been proved to deal with two forms of training biases effectively, and it is agnostic to a specific model or data type (Koh & Liang, 2017; Koh et al., 2019).

Inspired by the success of influence-based data resampling, in this paper, we would like to ask the following question: *what would happen if we relabel harmful training data based on influence analysis results*? Our motivations on performing data relabeling via influence analysis are twofold. (i) Relabeling noisy samples is able to prevent the model from memorizing the corrupted labels. (ii) Relabeling clean but biased samples is helpful to improve model's robustness to harmful samples. Despite the potential benefits of data relabeling, it is still challenging to develop an influence-based relabeling approach that has a theoretical guarantee on the model's performance improvement after training with relabeled data.

To answer the question, we first follow (Koh et al., 2019) to measure the influence of each training sample on model's predictions and identify the harmful training samples which would cause an increase in the test loss. Next, we investigate whether relabeling the identified harmful samples rather than discarding them can improve the test performance. To achieve this, we start from binary classification tasks where relabeling a training sample is to convert its binary label from $y$ to $1 - y$. We theoretically prove that relabeling harmful training data via influence analysis can achieve lower test loss than simply discarding them for binary classification. Furthermore, we design a novel relabeling function $\mathcal{R}$ for multi-class classification tasks and prove that the advantage of relabeling the identified harmful samples using $\mathcal{R}$ in reducing model's test loss still holds. Following the influence-based resampling approaches (Wang et al., 2018; Ting & Brochu, 2018; Ren et al., 2020; Wang et al., 2020b), we only use the test loss for theoretical analysis and empirically calculate influence function with a small but unbiased validation set by assuming the validation set is sampled from the same distribution as the test set. In this way, using the validation loss to calculate the influence function is an unbiased estimation of the true influence function. Otherwise, the problem may lie in the category of transfer learning which is beyond the scope of this work.

To summarize, this work makes the following contributions. **First**, we propose to combine influence functions with data relabeling for reducing training biases and we develop an end-to-end influence-based relabeling framework named RDIA that reuses harmful training samples toward better model performance. **Second**, we design a novel relabeling function $\mathcal{R}$ and theoretically prove that applying $\mathcal{R}$ over harmful training samples identified by influence functions is able to achieve lower test loss for any classification tasks using cross-entropy loss function. **Third**, we conduct extensive experiments on real datasets in different domains. The results demonstrate that (i) RDIA is effective in reusing harmful training samples towards higher model performance, surpassing the existing influence-based resampling approaches, and (ii) RDIA improves model's robustness to label noise, outperforming the current resampling methods by large margins.

## 2 BACKGROUND

Let $D = \{(x_i, y_i) \in \mathcal{X} \times \mathcal{Y} \mid 1 \leq i \leq N\}$ be the training set which are sampled from $P_{train}(x, y)$. Let $z_i = (x_i, y_i)$ where $x_i \in \mathbb{R}^d$ and $y_i \in \mathbb{R}^K$. Let $\varphi(x, \theta)$ be a model's prediction for $x$, where $\theta \in \mathbb{R}^p$ is the parameter set of the model. We denote the loss of sample $z_i$ by $l(z_i, \theta) = L(y_i, \varphi(x_i, \theta))$ and use $l_i(\theta)$ to represent $l(z_i, \theta)$. We consider the standard empirical risk minimization (ERM) as the optimization objective. Formally, the empirical risk over $D$ is defined as: $L(D; \theta) = \frac{1}{N} \sum_{i=1}^{N} l_i(\theta)$. Since our relabeling function is dependent to the loss function, we focus on the most effective and versatile loss, i.e., Cross Entropy loss for any classification tasks.

**Influence functions.** Influence functions, stemming from *Robust Statistics* (Huber, 2004), have provided an efficient way to estimate how a small perturbation of a training sample would change the model's predictions (Koh & Liang, 2017; Koh et al., 2019; Yu et al., 2020). Let $\hat{\theta} = \arg\min_\theta \frac{1}{N} \sum_{n=1}^{N} l_n(\theta)$ be the optimal model parameters on convergence. When upweighting a training sample $z_i$ on its loss term by an infinitesimal step $\epsilon_i$, we obtain the new optimal parameters $\hat{\theta}_{\epsilon_i}$ on convergence as : $\hat{\theta}_{\epsilon_i} = \arg\min_\theta \frac{1}{N} \sum_{n=1}^{N} l_n(\theta_i) + \epsilon_i l_i(\theta)$. Based on influence functions (Cook & Weisberg, 1980; Koh & Liang, 2017), we have the following closed-form expression to estimate

the *change in model parameters* when upweighting $z_i$ by $\epsilon_i$:

$$\psi_\theta(z_i) \triangleq \frac{\mathrm{d}\hat{\theta}_{\epsilon_i}}{\mathrm{d}\epsilon_i}|_{\epsilon_i=0} = -H_{\hat{\theta}}^{-1}\nabla_\theta l_i(\hat{\theta}), \tag{1}$$

where $H_{\hat{\theta}} \triangleq \frac{1}{N}\sum_{n=1}^{N}\nabla_\theta^2 l_n(\hat{\theta})$ is the Hessian matrix and $\nabla_\theta^2 l_n(\theta)$ is the second derivative of the loss at training point $z_n$ with respect to $\theta$. Using the chain rule, we can estimate the *change of model's prediction* at a test data $z_j^c$ sampled from the given test distribution $P_{test}$ (Koh & Liang, 2017) :

$$\Phi_\theta(z_i, z_j^c) \triangleq \frac{\mathrm{d}l_j(\hat{\theta}_{\epsilon_i})}{\mathrm{d}\epsilon_i}|_{\epsilon_i=0} = -\nabla_\theta l_j(\hat{\theta})H_{\hat{\theta}}^{-1}\nabla_\theta l_i(\hat{\theta}). \tag{2}$$

At a fine-grained level, we can measure the influence of perturbing training sample $z_i$ from $(x_i, y_i)$ to $(x_i, y_i + \delta)$. Let $z_{i\delta} = (x_i, y_i + \delta)$ and the new loss $l_i(z_{i\delta}, \theta) = L(y_i + \delta, \varphi(x_i, \theta))$. According to (Koh & Liang, 2017), the optimal parameters $\hat{\theta}_{\epsilon_i\delta_i}$ after performing perturbation on $z_i$ is $\hat{\theta}_{\epsilon_i\delta_i} = \arg\min_\theta \frac{1}{N}\sum_{n=1}^{N} l_n(\theta) + \epsilon_i l_i(z_{i\delta}, \theta) - \epsilon_i l_i(\theta)$. This allows us to estimate the *change in model parameters after the fine-grained data perturbation* using influence functions as:

$$\begin{aligned}\frac{\mathrm{d}\hat{\theta}_{\epsilon_i\delta_i}}{\mathrm{d}\epsilon_i}|_{\epsilon_i=0} &= \psi_\theta(z_{i\delta}) - \psi_\theta(z_i) \\ &= -H_{\hat{\theta}}^{-1}(\nabla_\theta l_i(z_{i\delta}, \hat{\theta}) - \nabla_\theta l_i(\hat{\theta})).\end{aligned} \tag{3}$$

Further, the *influence* of perturbing $z_i$ by $z_{i\delta}$ on model's prediction at test sample $z_j^c$ is the following:

$$\begin{aligned}\eta_{\theta\delta}(z_i, z_j^c) &\triangleq \frac{\mathrm{d}l_j(\hat{\theta}_{\epsilon_i\delta_i})}{\mathrm{d}\epsilon_i}|_{\epsilon_i=0} \\ &= -\nabla_\theta l_j(\hat{\theta})H_{\hat{\theta}}^{-1}(\nabla_\theta l_i(z_{i\delta}, \hat{\theta}) - \nabla_\theta l_i(\hat{\theta})).\end{aligned} \tag{4}$$

It is important to notice that Eq. (4) holds for arbitrary $\delta$ when $\epsilon_i$ is approaching 0. This provides the feasibility of measuring how relabeling a training samples could influence the model's predictions.

**Influence-based resampling approaches.** Previous researches (Koh & Liang, 2017; Wang et al., 2020b) have shown that influence functions have strong ability to identify harmful samples from the whole training set, which is agnostic to the specific model or data structure. Inspired by this, many influence-based resampling approaches (Ting & Brochu, 2018; Wang et al., 2018; 2020b) proposed to discard or downweight the identified harmful samples to reduce the test loss. However, different from previous works which focus on estimating the influence of each training sample on the test performance using Eq. (1)-(2), we perform the fine-grained perturbation on a training sample's label and evaluate its influence using Eq. (3)-(4). Further, our work tries to build an end-to-end influence-based relabeling framework to reuse the harmful samples with a theoretical guarantee on the final model performance for any classification tasks. To be specific, we demonstrate that *harmful training instances after being relabeled properly do make contributions to improve the final model performance*, which provides *a novel viewpoint on handling biased training data*.

## 3 METHODOLOGY

Assume we have $Q = \{(x_j^c, y_j^c) \in \mathcal{X} \times \mathcal{Y} \mid 1 \leq j \leq M\}$ sampled from the test distribution $P_{test}$ and our objective is to minimize the test risk $L(Q; \theta) = \frac{1}{M}\sum_{j=1}^{M} l_j^c(\theta)$. Due to the harmful training samples in $D$, the optimal $\hat{\theta}$ which minimizes the empirical risk over training set $D$ may not be the best risk minimizer over $Q$. To solve this issue, we propose a novel data relabeling framework named RDIA which aims to identify and reuse harmful training instances towards better model performance. We design a relabeling function $\mathcal{R}$ that allows the model to achieve lower test risk after being trained with the relabeled harmful instances for any classification tasks. In what follows, we first give an overview of the RDIA framework. Then we describe the details of the major steps in RDIA and provide theoretical analysis on how the relabeled harmful samples are useful to further reduce the test risk. The algorithm of RDIA could be found in Appendix A

### 3.1 OVERVIEW OF RDIA

Figure 1 provides an overview of RDIA, which consists of four major steps: *Model training*, *Harmful samples identification*, *Relabeling harmful samples via influence analysis* and *Model retraining*.

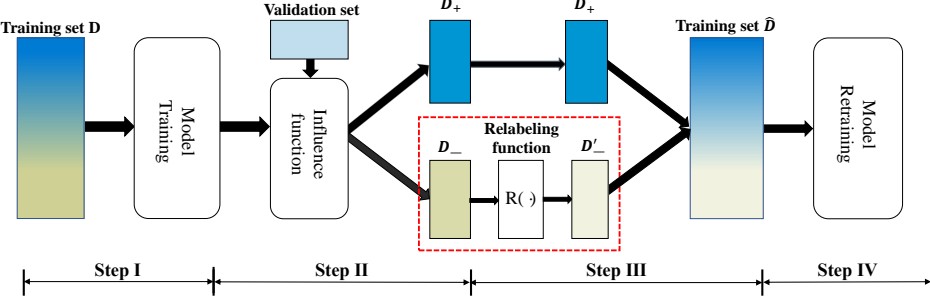

Figure 1: The overview of RDIA. We devise relabeling function $\mathcal{R}$ to change the labels of the identified harmful training samples in $D_-$.

**Step I:** *Model training* is to train a model based on the training set $D$ until convergence and get the model parameters $\hat{\theta}$.

**Step II:** *Harmful samples identification* is to compute the influence of perturbing the loss term of each training sample $z_i \in D$ on test risk using Eq. (2) and then use it to identify the harmful training samples from $D$. We denote the set of identified harmful training samples as $D_-$ and the set of remaining training instances as $D_+$. The details of this step are provided in Section 3.2.

**Step III:** *Relabeling harmful samples via influence analysis* is to apply the relabeling function to modify the label of each identified harmful training sample in $D_-$ and obtain the set of relabeled harmful training samples denoted as $D'_-$. We introduce our relabeling function $\mathcal{R}$ and theoretically prove that updating the model's parameters with new training set $\hat{D} = D'_- \cup D_+$ can achieve lower test risk over $Q$ than simply discarding or downweighting $D_-$ in Section 3.3.

**Step IV**: *Model retraining* is to retrain the model using $\hat{D}$ till convergence to get the final optimal parameters $\hat{\theta}_{\epsilon\mathcal{R}}$.

### 3.2 HARMFUL SAMPLES IDENTIFICATION

In the second step, we compute $D_- \subseteq D$ which contains harmful training samples from the original training set $D$. Intuitively, a training sample is harmful to the model performance if removing it from the training set would reduce the test risk over $Q$. Based on influence functions, we can measure one sample's influence on test risk without prohibitive leave-one-out training. According to Eq. (1)-(2), if we add a small perturbation $\epsilon_i$ on the loss term of $z_i$ to change its weight, the *change of test loss* at a test sample $z_j^c \in Q$ can be estimated as follows:

$$l(z_j^c, \hat{\theta}_{\epsilon_i}) - l(z_j^c, \hat{\theta}) \approx \epsilon_i \times \Phi_\theta(z_i, z_j^c), \tag{5}$$

where $\Phi_\theta(\cdot, \cdot)$ is computed by Eq. (2). We then estimate the influence of perturbing $z_i$ on the whole test risk as follows:

$$l(Q, \hat{\theta}_{\epsilon_i}) - l(Q, \hat{\theta}) \approx \epsilon_i \times \sum_{j=1}^{M} \Phi_\theta(z_i, z_j^c). \tag{6}$$

Henceforth, we denote by $\Phi_\theta(z_i) = \sum_{j=1}^{M} \Phi_\theta(z_i, z_j^c)$ the influence of perturbing the loss term of $z_i$ on the test risk over $Q$. It is worth mentioning that given $\epsilon_i \in [-\frac{1}{N}, 0)$, Eq. (6) computes the influence of downweighting or discarding the training sample $z_i$. We denote $D_- = \{z_i \in D \mid \Phi_\theta(z_i) > 0\}$ as harmful training samples. Similar to (Wang et al., 2020b), we assume that each training sample influences the test risk independently. We derive the Lemma 1.

**Lemma 1.** *Discarding or downweighting the training samples in $D_- = \{z_i \in D \mid \Phi_\theta(z_i) > 0\}$ from $D$ could lead to a model with lower test risk over $Q$:*

$$L(Q, \hat{\theta}_\epsilon) - L(Q, \hat{\theta}) \approx -\frac{1}{N} \sum_{z_i \in D_-} \Phi_\theta(z_i) \leq 0, \tag{7}$$

*where $\hat{\theta}_\epsilon$ denotes the optimal model parameters obtained by updating the model's parameters with discarding or downweighting samples in $D_-$.*

Lemma 1 explains why influence-based resampling approaches have strong ability to resolve training biases and the proof of Lemma 1 is provided in Appendix B. In practice, to further tolerate the estimation error in $\Phi_\theta(z_i)$ which may result in the wrong identification of harmful training samples, we select $D_- = \{z_i \in D \mid \Phi_\theta(z_i) > \alpha\}$ where the hyperparameter $\alpha$ controls the proportion of harmful samples to be relabeled eventually. We conduct the experiment to show the effects of $\alpha$ and the validation set in Section 5.3.

## 3.3 RELABELING HARMFUL SAMPLES VIA INFLUENCE ANALYSIS

In the third step, we propose a relabeling function $\mathcal{R}$ and ensure the test risk would be reduced after training with the relabeled harmful samples $D'_-$. To achieve this, we start from a special case (i.e., binary classification) and then extend to any classification tasks.

**Relabeling harmful instances on binary classification.** We start with binary classification where the relabeling function $R$ is straightforward. Since the label set $\mathcal{Y}$ is $\{0, 1\}$, we have $R(z) = 1 - y$ for any $z = (x, y) \in D$. Recall that $\varphi(x_i, \theta)$ denotes the model output for $x_i$ and the training loss of $z_i = (x_i, y_i) \in D$ is: $l_i(\theta) = -y_i \log(\varphi(x_i, \theta)) - (1 - y_i) \log(1 - \varphi(x_i, \theta))$

To compute the influence of relabeling a training sample $z_i$ in $D$, we first consider the case that $y_i = 1$ and $R(z_i) = 0$. The loss $l_i(\theta)$ at $z_i$ is changed from $-\log(\varphi(x_i, \theta))$ to $-\log(1 - \varphi(x_i, \theta))$. Letting $z_{iR} = (x_i, 1 - y_i)$ and $w(z_i, \theta) = \nabla_\theta l_i(z_{iR}, \theta) - \nabla_\theta l_i(\theta)$, we have:

$$w(z_i, \theta) = -\nabla_\theta \log(1 - \varphi(x_i, \theta)) + \nabla_\theta \log(\varphi(x_i, \theta)) = \frac{-\nabla_\theta l_i(\theta)}{1 - \varphi(x_i, \theta)}. \tag{8}$$

According to Eq. (2),(4) and (8), the influence of relabeling $z_i$ on model's prediction at test sample $z_j^c$ is:

$$\eta_{\theta R}(z_i, z_j^c) = -\nabla_\theta l_j(\hat{\theta}) H_{\hat{\theta}}^{-1} w(z_i, \hat{\theta}) = -\nabla_\theta l_j(\hat{\theta}) H_{\hat{\theta}}^{-1} \left( -\frac{\nabla_\theta l_i(\hat{\theta})}{1 - \varphi(x_i, \hat{\theta})} \right) = \frac{-\Phi_\theta(z_i, z_j^c)}{1 - \varphi(x_i, \hat{\theta})}. \tag{9}$$

Similarly, when the label $y_i$ in $z_i$ is 0 and $R(z_i) = 1$, we can derive the influence of relabeling $z_i$ at $z_j^c$ as $\eta_{\theta R}(z_i, z_j^c) = \frac{-\Phi_\theta(z_i, z_j^c)}{\varphi(x_i, \hat{\theta})}$. Let $\hat{\theta}_{\epsilon_i R_i}$ denote the optimal parameters after relabeling $z_i$. Similar to Eq. (6), we could extend the influence of relabeling $z_i$ at $z_j^c$ to the whole test risk over $Q$ as:

$$l(Q, \hat{\theta}_{\epsilon_i R_i}) - l(Q, \hat{\theta}) \approx \epsilon_i \times \sum_{j=1}^{M} \eta_{\theta R}(z_i, z_j^c). \tag{10}$$

According to Eq. (9), the influence of relabeling training samples $\eta_{\theta R}(z_i, z_j^c)$ is related to the influence of perturbing the loss term of training samples, i.e., $\Phi_\theta(z_i, z_j^c)$. In this way, the change of test risk by relabeling $z_i$ (Eq. (10)) and that by perturbing $z_i$ (Eq. (6)) are interrelated. Then we derive the Theorem 1 and the proof can be found in Appendix B.

**Theorem 1.** *In binary classification, let $\sigma$ be the infimum of $\frac{\varphi(x_i, \hat{\theta})}{1 - \varphi(x_i, \hat{\theta})}$ and $\frac{1 - \varphi(x_i, \hat{\theta})}{\varphi(x_i, \hat{\theta})}$ and $D_- = \{z_i \in D \mid \Phi_\theta(z_i) > 0\}$. Relabeling the samples in $D_-$ can achieve lower test risk than discarding or downweighting them from $D$, because the following inequality holds.*

$$L(Q, \hat{\theta}_{\epsilon R}) - L(Q, \hat{\theta}_\epsilon) \approx -\frac{\sigma}{N} \sum_{z_i \in D_-} \Phi_\theta(z_i) \leq 0. \tag{11}$$

Theorem 1 shows that relabeling the samples in $D_-$ could achieve lower test risk than simply discarding or downweighting $D_-$ from the training set for binary classification tasks. We then provide some intuitions on the benefits of relabeling harmful training samples. In the context of binary classification, if a training sample $z$ in $D_-$ is noisy, our relabeling method corrects the label noise and improve training data quality; otherwise, $z$ is very likely to be biased due to its negative impact on the test risk, and relabeling it might improve model's robustness.

**Relabeling harmful instances on any classification tasks.** We now introduce a relabeling function $\mathcal{R}$ that can be used for any classification tasks. For a classification problem with $K$ class labels ($K \geq 2$), we represent each label $y$ as a $K$-dimensional one-hot vector. The CE loss at $z_i$ is: $l_i(\theta) = -\sum_{k=1}^{K} y_{i,k} \log(\varphi_k(x_i, \theta))$. Intuitively, the proposed relabeling function $\mathcal{R}$ should satisfy the following principles:

- **Consistency**: $\mathcal{R}$ should produce a $K$-dimensional label vector: $\mathcal{R}(x_i, y_i) = y_i' \in [0, 1]^K$.
- **Effectiveness**: applying $\mathcal{R}$ over harmful training samples $D_-$ should guarantee the resultant test risk is no larger than the one achieved by simply discarding them.

For the consistency principle, we require the new label $y_i'$ to be $K$-dimensional where $y_{ik}'$ describes the likelihood that $x_i$ takes the $k$-th class label, $k \in [1, K]$. Here we do not require $\sum_{k=1}^{K} y_{ik}'$ to be one, because we focus on leveraging the identified harmful training samples towards better model performance instead of finding their truth labels.

Consider a training sample $z_i = (x_i, y_i)$ belonging to the $m$-th class ($m \in [1, K]$), i.e., $y_{i,m} = 1$. Let $\mathcal{R}(x_i, y_i) = y_i'$, we propose the following relabeling function $\mathcal{R}$ that fulfills the above two principles:

$$y_{i,k}' = \begin{cases} 0, & \text{if } k = m \\ \log_{\varphi_k} \sqrt[K-1]{1 - \varphi_m}, & \text{otherwise} \end{cases}, \tag{12}$$

where $\varphi(x_i, \hat{\theta}) = (\varphi_1, \cdots, \varphi_K)$ is the probability distribution over $K$ classes produced by the model with parameters $\hat{\theta}$, i.e., $\varphi_i \in [0, 1]$ and $\sum_{i=1}^{K} \varphi_i = 1$. It is easy to check that our proposed relabeling function $\mathcal{R}$ in Eq. (12) satisfies the first principle. Interestingly, we can verify that for $K = 2$, we have $\mathcal{R}(z_i) = 1 - y_i$. We further prove the effectiveness of $\mathcal{R}$ using the Lemma 2.

**Lemma 2.** *When applying the relabeling function $\mathcal{R}$ in Eq. (12) over a training sample $z_i \in D$ with a class label $m$, the CE loss $l_i(\theta)$ at $z_i$ is changed from $-\log(\varphi_m(x_i, \theta))$ to $-\log(1 - \varphi_m(x_i, \theta))$.*

It is interesting to verify the change in loss $l_i(\theta)$ acts as an extension of the binary classification. Similar to Theorem 1, we can drive the following theorem using Eq. (9)-(10).

**Theorem 2.** *In multi-class classification, let $\varphi_y(x_i, \hat{\theta})$ denote the probability that $z_i$ is classified as its truth class label by the model with the optimal parameters $\hat{\theta}$ on $D$, and $\sigma$ be the infimum of $\frac{\varphi_y(x_i, \hat{\theta})}{1 - \varphi_y(x_i, \hat{\theta})}$. Relabeling the samples in $D_- = \{z_i \in D \mid \Phi_\theta(z_i) > 0\}$ with $\mathcal{R}$ leads to a test risk lower than the one achieved by discarding or downweighting $D_-$. Formally, we have:*

$$L(Q, \hat{\theta}_{\epsilon\mathcal{R}}) - L(Q, \hat{\theta}_\epsilon) \approx -\frac{\sigma}{N} \sum_{z_i \in D_-} \Phi_\theta(z_i) \leq 0. \tag{13}$$

Theorem 2 shows that using our proposed relabeling $\mathcal{R}$ can further reduce the test risk than simply discarding or downweighting $D_-$ from the training set for any classification tasks. The detailed proofs of Lemma 2 and Theorem 2 are provided in Appendix B.

## 4 DISCUSSIONS

In this section, we provide numerical analysis on the superior performance of RDIA against other influence-based resampling approaches (Wang et al., 2018; 2020b). Then we discuss the extension of RDIA in exploiting training loss information.

**Numerical analysis.** Consider a training point $z_i \in D_-$ belonging to class $m$, where $D_-$ is specified in Section 3.2. According to Eq. (13), if we use $\mathcal{R}$ to assign $z_i$ with a new label $y_i' = \mathcal{R}(z_i)$ instead of discarding or downweighting $z_i$, the difference in the test risk over $Q$ can be computed as:

$$g(z_i) = l_i(Q, \hat{\theta}_{\epsilon_i \mathcal{R}_i}) - l_i(Q, \hat{\theta}_{\epsilon_i}) \approx -\frac{1}{N} \times \frac{\varphi_m(x_i, \hat{\theta})}{1 - \varphi_m(x_i, \hat{\theta})} \Phi_\theta(z_i). \tag{14}$$

$z_i \in D_-$ means the $\Phi_\theta(z_i)$ in Eq. (14) is positive, and hence we have $g(z_i) < 0$. If the model's prediction for $z_i$ with the optimal parameters $\hat{\theta}$ is correct, $\varphi_m(x_i, \hat{\theta})$ is the largest component in the vector $\varphi(x_i, \hat{\theta})$. We consider $z_i$ as a *more harmful* sample because it has negative influence on the test loss yet the model has learnt some features from $z_i$ that connects $x_i$ to class $m$. In practice, $z_i$ is very likely to be a noisy or biased training sample. Interestingly, from Eq. (14), we can see that a small increase in $\varphi_m(x_i, \hat{\theta})$ will lead to a rapid increase in $g(z_i)$. This indicates relabeling such *more harmful* training samples leads to significant performance gain.

**Extension of RDIA.** In practice, due to the complexity of calculating the influence functions, identifying harmful samples via influence analysis could incur high computational cost, especially when

training complex models like deep neural networks. To address the problems, we further extend RDIA by using training loss to identify harmful samples for deep models, and we dub this extension as RDIA-LS. We empirically show that RDIA-LS is effective and efficient to handle training data with corrupted labels for deep learning, which spotlights the great scalability of our approach. The details of RDIA-LS are provided in Appendix F.

## 5 EXPERIMENTS

In this section, we conduct experiments to evaluate the effectiveness and robustness of our RDIA. We also perform the ablation study to show how hyperparameter $\alpha$ and the size of validation set affect the performance of RDIA. The visualization of identified harmful samples and comparison with other loss-based approaches are provided in Appendix D and Appendix G.

### 5.1 EXPERIMENTAL SETTINGS

**Datasets.** To verify the effectiveness of RDIA, we perform extensive experiments on ten public data sets from different domains, including NLP, CV, CTR, etc. Since all the datasets are clean, we build a noise transition matrix $P = \{P_{ij}\}_{K \times K}$ to verify the robustness of our proposed approaches for combating noisy labels, where $K$ denotes the class number and $P_{ij}$ denotes the probability of a clean label $i$ being flipped to a noisy label $j$. In our experiment, we use the noise ratio $\tau$ to determine the rate of how many labels are manually corrupted and each clean label has the same probability of being flipped to other classes, i.e., $P_{ij} = \frac{\tau}{K-1}$. More details about the statistics of the datasets and Tr-Va-Te divisions are provided in Appendix C.

**Comparison methods.** We compared our proposed relabeling method **RDIA** with the following baselines, all of which are agnostic of specific model or data structure. (1) **ERM**: it means training a model with all the training data with the cross-entropy loss. (2) **Random**: it is a basic relabeling method that randomly selects and changes the label of training samples. (3) **OptLR** (Ting & Brochu, 2018): it is a weighted sampling method which assigns each training sample with a weight proportional to its impact on the change in model's parameters $\psi_\theta$. Specifically, the weight of $z_i$ is $\max\{\alpha, \min\{1, \lambda\psi_\theta(z_i)\}\}$. We set $\alpha$ and $\lambda$ to be $1/\max\{\psi_\theta(z_i)\}$ and $1/\max\{\Phi_\theta(z_i)\}$, respectively. (4) **Dropout** (Wang et al., 2018): it is an unweighted subsampling method which simply discards $D_-$ from the training set, i.e., removing all training data with negative influence on the test loss. (5) **UIDS** (Wang et al., 2020b): it it is an unweighted subsampling method which uses Linear sampling method or Sigmoid sampling method to resample the training data based on influence functions $\Phi_\theta(z_i)$. It is the best-performing method among all the existing influence-based methods.

We implemented all the comparison methods by using their published source codes in Pytorch and ran all the experiments on a server with 2 Intel Xeon 1.7GHz CPUs, 128 GB of RAM and a single NVIDIA 2080 Ti GPU. All the baselines are tuned with clean validation data for best model performance. To measure the performance of all the approaches, we followed (Wang et al., 2020b) and used the test loss as the metric since we aim to optimize the test loss via influence analysis.

**Implementation details.** For each of the ten datasets, we adopted logistic regression (convex optimization) as the binary classifier (for MNIST and CIFAR10, we randomly choose two classes to perform binary classification). As for multi-class classification, we implemented two deep models (non-convex optimization), LeNet (2 convolutional layers and 1 fully connected layers) and a CNN with 6 convolutional layers followed by 2 fully connected layers used in (Wang et al., 2019) on MNIST and CIFAR10. The hyperparameter $\alpha$ is also tuned in [0, 0.001, 0.002, ...,0.01] with the clean validation set for best performance. More detailed settings are provided in Appendix C.

### 5.2 EXPERIMENTAL RESULTS

**Effectiveness of RDIA.** To verify the effectiveness of RDIA , we conduct experiments on 10 clean datasets with three different models. The experiments are repeated 5 times and the averaged test loss with standard deviation results are reported in Table 1 and Table 2.We have the following important observations.

First, our proposed RDIA yields the lowest test loss over 9 out of 10 datasets using logistic regression. It outperforms ERM on all the datasets, which indicates the effectiveness of relabeling

Table 1: Performance comparison results with logistic regression on binary classification task. Average test loss (±std) over 5 repetitions are reported.

| Dataset | ERM | Random | OptLR | Dropout | UIDS | RDIA |
|---|---|---|---|---|---|---|
| Breast-cancer | 0.0914 | 0.2619 ± 0.0102 | 0.0934 ± 0.0015 | 0.0731 ± 0.0014 | 0.0786 ± 0.0006 | **0.0649 ± 0.0001** |
| Diabetes | 0.5170 | 0.5461 ± 0.0006 | 0.5232 ± 0.0012 | 0.5083 ± 0.0008 | 0.5068 ± 0.0004 | **0.4920 ± 0.0002** |
| News20 | 0.5157 | 0.5247 ± 0.0028 | 0.5253 ± 0.0021 | 0.5072 ± 0.0019 | 0.5075 ± 0.0012 | **0.5007 ± 0.0015** |
| Adult | 0.3383 | **0.3381 ± 0.0001** | 0.3547 ± 0.0001 | 0.3383 ± 0.0001 | 0.3383 ± 0.0001 | 0.3383 ± 0.0001 |
| Real-sim | 0.2606 | 0.2638 ± 0.0025 | 0.2884 ± 0.0151 | 0.2605 ± 0.0024 | 0.2607 ± 0.0031 | **0.2575 ± 0.0021** |
| Criteo1% | 0.4911 | 0.4919 ± 0.0011 | 0.4914 ± 0.0007 | 0.4995 ± 0.0025 | 0.4895 ± 0.0012 | **0.4894 ± 0.0010** |
| Covtype | 0.6936 | 0.6906 ± 0.0029 | 0.6907 ± 0.0026 | 0.6843 ± 0.0023 | 0.6784 ± 0.0032 | **0.6776 ± 0.0024** |
| Avazu | 0.3449 | 0.3449 ± 0.0002 | 0.3450 ± 0.0002 | 0.3576 ± 0.0001 | 0.3447 ± 0.0001 | **0.3447 ± 0.0001** |
| MNIST | 0.0245 | 0.2543 ± 0.0005 | 0.0239 ± 0.0004 | 0.0221 ± 0.0002 | 0.0238 ± 0.0003 | **0.0207 ± 0.0001** |
| CIFAR10 | 0.5952 | 0.6174 ± 0.0025 | 0.6163 ± 0.0021 | 0.5946 ± 0.0017 | 0.5845 ± 0.0015 | **0.5806 ± 0.0012** |

Table 2: Performance comparison results with deep models on multi-classification task. Average test loss (±std) over 5 repetitions are reported.

| Dataset | ERM | Random | OptLR | Dropout | UIDS | RDIA |
|---|---|---|---|---|---|---|
| MNIST(LeNet) | 0.0283 | 0.0407 ± 0.0025 | 0.0756 ± 0.0102 | 0.0256 ± 0.0002 | 0.0261 ± 0.0002 | **0.0251 ± 0.0005** |
| MNIST(CNN) | 0.0322 | 0.0385 ± 0.0003 | 0.0576 ± 0.0042 | 0.0289 ± 0.0002 | 0.0302 ± 0.0011 | **0.0281 ± 0.0006** |
| CIFAR10(LeNet) | 1.1641 | 1.6247 ± 0.0223 | 1.8341 ± 0.0421 | 1.1721 ± 0.0019 | **1.1534 ± 0.0017** | 1.2631 ± 0.0015 |
| CIFAR10(CNN) | 0.7744 | 0.8303 ± 0.0162 | 1.2303 ± 0.0329 | 0.7859 ± 0.0016 | 0.7910 ± 0.0013 | **0.6052 ± 0.0029** |

training samples via influence functions to resolve training biases. Furthermore, RDIA outperforms the state-of-the-art resampling method UIDS on all the datasets except Avazu, which indicates the effectiveness of reusing harmful training samples via relabeling towards higher model performance.

Second, when training deep models, RDIA achieves the best test loss on MNIST+LeNet, MNIST+CNN, and CIFAR10+CNN, where it outperforms UIDS by a large margin. We observe LeNet performs much worse than CNN on CIFAR10 using the original training set (i.e., the results of ERM) due to its simple architecture. Note that the poor classification results for clean and unbiased training data would interfere the identification of true harmful training samples. Hence, RDIA performs similarly to Random which introduces random noises into the training set and the performance suffers. But we want to emphasize that when training a more suitable model (e.g., CNN) on CIFAR10, RDIA is more effective to improve model's performance.

Third, Random performs worse than ERM on all the cases except on Adult. This indicates that randomly relabeling harmful training samples may easily inject noisy data that hurt model's performance significantly. In contrast, our proposed relabeling function is effective to assign appropriate labels to harmful training samples that benefit the test loss.

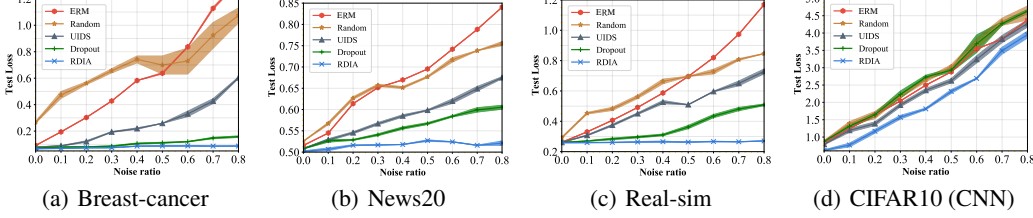

(a) Breast-cancer     (b) News20     (c) Real-sim     (d) CIFAR10 (CNN)

Figure 2: Test loss results with different noise ratios. Shaded regions indicate standard deviation.

**Robustness to label noises.** In order to investigate the robustness of RDIA to noise labels, we set the noise ratio $\tau$ from 0 to 0.8 to manually corrupt the labels in the four datasets from different domains, while the results on the other datasets have similar trends. Figure 2 reports the average test loss of all the influence-based approaches on four noisy datasets with different noise ratios. First, thanks to the high accuracy of estimating the influence functions on logistic regression, all influence-based approaches consistently outperform ERM, which indicates the effectiveness of using influence functions to identify noisy samples. Figure 2(a), 2(b) and 2(c) show that RDIA performs significantly better than the other influence-based approaches. As the noise ratio becomes larger, the test loss of all the other approaches increases while the test loss reported by RDIA is generally unchanged. This verifies the robustness of RDIAto high noise ratios. We surprisingly find that RDIA at 0.8 noise ratio achieves lower test loss than ERM at zero noise ratio. The reason might be that RDIA could leverage all the training samples and fix noisy labels properly to boost the per-

Table 3: Effect of hyperparameter $\alpha$ on RDIA (11684 training samples in total).

| | Noise ratio | 0 | 0.2 | 0.5 | 0.8 |
|---|---|---|---|---|---|
| ERM | Relabeling number | 0 | 0 | 0 | 0 |
| | Test loss | 0.0245 | 0.2567 | 0.6942 | 1.5975 |
| $\alpha = 0.01$ | Relabeling number | 6 | 1439 | 3626 | 6140 |
| | Test loss | 0.0235 | 0.0443 | 0.0903 | 0.1009 |
| $\alpha = 0.002$ | Relabeling number | 71 | 1721 | 4139 | 6545 |
| | Test loss | **0.0207** | **0.0315** | 0.0519 | 0.0465 |
| $\alpha = 0.0002$ | Relabeling number | 530 | 1804 | 4193 | 6589 |
| | Test loss | 0.0903 | 0.0392 | **0.0410** | **0.0405** |

Table 4: Effect of the number of validation samples used in RDIA (11684 training samples in total).

| | Number of validation samples | 100 | 200 | 500 | 1000 |
|---|---|---|---|---|---|
| ERM | Validation loss | 0.5337 | 0.5309 | 0.5233 | 0.5275 |
| | Test loss | | 0.5219 | | |
| UIDS | Validation loss | 0.2388 | 0.2269 | 0.2417 | 0.2331 |
| | Test loss | 0.4928 | 0.3873 | 0.2783 | 0.2409 |
| RDIA | Validation loss | 0.0679 | 0.0583 | **0.0494** | 0.0514 |
| | Test loss | 0.3430 | 0.2080 | 0.1396 | **0.0847** |

formance. Second, Figure 2(d) shows that when combating noisy labels for deep models, RDIA still suffers from the noisy labels like other baselines because the estimation of influence functions with deep models is not accurate enough to filter out all noisy labels. However, RDIA could still relabel the most negative samples to reduce the test loss.

## 5.3 ABLATION STUDY

Finally, we investigate the effect of different values of hyperparameters $\alpha$ and size of validation set on the performance of RDIA using MNIST with logistic regression.

**Hyparameter $\alpha$.** As discussed in Section 3.3, by varying $\alpha$, we can derive the percentage of relabeled training data against the complete training set in RDIA. Table 3 provides the results of how many samples are relabeled and how test loss is changed with different values of $\alpha$ under different noise ratios. First, when noise ratio equals to 0, there are few biased samples in the training set. In this case, simply relabeling all the identified harmful samples will hurt the performance while using relatively larger $\alpha$ could report lower test loss. Second, when noise ratio is 0.8, RDIA achieves better performance with smaller $\alpha$. This is reasonable since most of training samples involve label noises and increasing $\alpha$ facilitates the relabeling of noisy samples.

**Size of the validation set.** As discussed in Section 3.3, we use the validation set instead of the test set to estimate the influence of each training sample. Table 4 shows the results of how the number of validation samples affects the model performance. We conduct the experiments under 40% noise rates and find the optimal hyperparameter $\alpha \in [0.0002, 0.01]$ to get the best results of RDIA.

We have the following observations. 1) Using only 100 validation samples with RDIA achieves 35% lower test loss than ERM. 2) As the number of validation samples increases, RDIA significantly outperforms ERM, achieving up to 90% relative lower in test loss. The reason is that, as the number of validation sets increases, the validation set can gradually reflect the true distribution of test data. In this way, the estimated influence functions are accurate enough to filter out most harmful training samples for the test set. 3) RDIA consistently outperforms UIDS with different sizes of validation set, which empirically shows the effectiveness of our relabeling function $\mathcal{R}$.

## 6 CONCLUSION

In this paper, we propose to perform data relabeling based on influence functions to resolve the training bias issue. We develop a novel relabeling framework named RDIA, which reuses the information of harmful training samples identified by influence analysis towards higher model performance. We theoretically prove that RDIA can further reduce the test loss than simply discarding harmful training samples on any classification tasks using the cross-entropy loss function. Extensive experiments on real datasets verify the effectiveness of RDIA in enhancing model's robustness and final performance, compared with various resampling and relabeling techniques.

**Reproducibility:** We clarify the assumptions in Section 2 and provide the complete proofs of Lemmas, Theorems in Appendix B. The statistics of datasets, the data processing, and the details of the experimental settings are described in Appendix C. Our code could be found in the `https://github.com/Viperccc/RDIA`.

## ACKNOWLEDGMENT

This work is supported by Shanghai Municipal Science and Technology Major Project (2021SHZDZX0102), the Tencent Wechat Rhino-Bird Focused Research Program, and SJTU Global Strategic Partnership Fund (2021 SJTU-HKUST). Yanyan Shen is the corresponding author of this paper.

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

# Appendix

In this appendix, we first provide the algorithm of RDIA (Appendix A) and the complete proofs of the Lemmas and Theorems (Appendix B) in the main text. Then we give the details of the experimental settings (Appendix C), the extensive analysis of our approach (Appendix D), and the visualization of identified harmful samples (Appendix E). We then describe RDIA-LS, an extension of RDIA, to spotlight the scalability of our approach RDIA (Appendix F) and provide empirical results to show that RDIA-LS is effective and efficient to handle training data with corrupted labels for deep learning (Appendix G). Finally, we provide additional discussions about the existing data relabeling approaches (Appendix H)

## A    RDIA ALGORITHM

---
**Algorithm 1:** RDIA

---
**Input:** Training model $\theta$, biased training set $D = \{(x_i, y_i)\}_{i=1}^N$, learning rate $\beta$, sample
selection ratio $\alpha$ such that $0 \leq \alpha \leq 1$, small and unbaised set $Q = \{(x_j^c, y_j^c)\}_{i=1}^M$

1   **Train** the model $\theta$ with $D$ until convergence to get $\hat{\theta}$;
2   **Initialize** $D_-, D_+ = \emptyset$
3   **for** $i \in [1, \ldots, N]$ **do**
4     **Calculate** the influence of the training sample $z_i = (x_i, y_i)$ on $Q$ using Eq. (6):
5     **if** $\Phi_\theta(z_i) > \alpha$ **then**
6       **Relabel** the identified harmful training samples with $z_i' \leftarrow \mathcal{R}(z_i)$;
7       $D_- \leftarrow D_- \cup \{z_i'\}$
8     **else if** $\Phi_\theta(z_i) < 0$ **then**
9       $D_+ \leftarrow D_+ \cup \{z_i\}$
10    **end**
11 **end**
12 **Obtain** the new training set $\hat{D} \leftarrow D_- \cup D_+$
13 **Retrain** the model with $\hat{D}$ till convergence to get the final model parameters $\hat{\theta}_{\epsilon\mathcal{R}}$

---

## B    PROOFS FOR LEMMAS AND THEOREMS

### B.1    PROOF OF LEMMA 1

Assume the perturbation $\epsilon_i$ on $z_i$ is infinitesimal and the influence of each training sample on the test risk is independent.

**Lemma 1.** *Discarding or downweighting the training samples in* $D_- = \{z_i \in D \mid \Phi_\theta(z_i) > 0\}$ *from $D$ could lead to a model with lower test risk over $Q$:*

$$L(Q, \hat{\theta}_\epsilon) - L(Q, \hat{\theta}) \approx -\frac{1}{N} \sum_{z_i \in D_-} \Phi_\theta(z_i) \leq 0,$$

*where $\hat{\theta}_\epsilon$ denotes the optimal model parameters obtained by updating the model's parameters with discarding or downweighting samples in $D_-$.*

*Proof.* Recall that $\hat{\theta}_{\epsilon_i} = \arg\min_\theta \frac{1}{N} \sum_{n=1}^N l_n(\theta_i) + \epsilon_i l_i(\theta)$. In this way, downweighting the training sample $z_i$ in $D_-$ means setting $\epsilon_i \in [-\frac{1}{N}, 0)$ (Noticed that $\epsilon_i = -\frac{1}{N}$ means discarding training sample $z_i$). For convenience of analysis, we set all $\epsilon_i$ equal to $-\frac{1}{N}$ and have $\Phi_\theta(z_i) \triangleq \sum_{j=1}^M \Phi_\theta(z_i, z_j^c)$. According to Eq. (6), we can estimate how the test risk is changed by discarding or downweighting

$z_i \in D_-$ as follows:

$$L(Q, \hat{\theta}_\epsilon) - L(Q, \hat{\theta}) = \sum_{z_i \in D_-} \sum_{j=1}^{M} l(z_j^c, \hat{\theta}_{\epsilon_i}) - l(z_j^c, \hat{\theta})$$

$$\approx \sum_{z_i \in D_-} \epsilon_i \times \sum_{i=1}^{M} \Phi_\theta(z_i, z_j^c)$$

$$= -\frac{1}{N} \sum_{z_i \in D_-} \Phi_\theta(z_i) \le 0$$

$\square$

## B.2 PROOF OF THEOREM 1

**Theorem 1.** *In binary classification, let $\sigma$ be the infimum of $\frac{\varphi(x_i, \hat{\theta})}{1 - \varphi(x_i, \hat{\theta})}$ and $\frac{1 - \varphi(x_i, \hat{\theta})}{\varphi(x_i, \hat{\theta})}$ and $D_- = \{z_i \in D \mid \Phi_\theta(z_i) > 0\}$. Relabeling the samples in $D_-$ can achieve lower test risk than discarding or downweighting them from $D$, because the following inequality holds.*

$$L(Q, \hat{\theta}_{\epsilon R}) - L(Q, \hat{\theta}_\epsilon) \approx -\frac{\sigma}{N} \sum_{z_i \in D_-} \Phi_\theta(z_i) \le 0.$$

*Proof.* Based on Eq. (9), we have:

$$\frac{\eta_{\theta R}(z_i, z_j^c)}{\Phi_\theta(z_i, z_j^c)} + 1 = \begin{cases} \dfrac{1 - \varphi(x_i, \hat{\theta})}{-\varphi(x_i, \hat{\theta})}, & \text{if } y_i = 0 \\[3mm] \dfrac{-\varphi(x_i, \hat{\theta})}{1 - \varphi(x_i, \hat{\theta})}, & \text{if } y_i = 1 \end{cases}$$

It is worth mentioning that $\hat{\theta}_{\epsilon_i R_i} = \arg\min_\theta \frac{1}{N} \sum_{n=1}^{N} l_n(\theta) + \epsilon_i l_i(z_{iR}, \theta) - \epsilon_i l_i(\theta)$. In this way, relabeling the training sample $z_i$ in $D_-$ means setting $\epsilon_i = \frac{1}{N}$.

Similar to the proof of Lemma 1, according to Eq. (6) and Eq. (10), we have:

$$L(Q, \hat{\theta}_{\epsilon R}) - L(Q, \hat{\theta}_\epsilon) = L(Q, \hat{\theta}_{\epsilon R}) - L(Q, \hat{\theta}) + L(Q, \hat{\theta}) - L(Q, \hat{\theta}_\epsilon)$$

$$= \sum_{z_i \in D_-} \sum_{j=1}^{M} l(z_j^c, \hat{\theta}_{\epsilon_i R_i}) - l(z_j^c, \hat{\theta}) - (l(z_j^c, \hat{\theta}_{\epsilon_i}) - l(z_j^c, \hat{\theta}))$$

$$\le \sum_{z_i \in D_-} \left( \sum_{j=1}^{M} \frac{1}{N} \eta_{\theta R}(z_i, z_j^c) + \frac{1}{N} \sum_{j=1}^{M} \Phi_\theta(z_i, z_j^c) \right)$$

$$= \frac{1}{N} \sum_{z_i \in D_-} \sum_{j=1}^{M} \left( \frac{\eta_\theta(z_i, z_j^c)}{\Phi_\theta(z_i, z_j^c)} + 1 \right) \Phi_\theta(z_i, z_j^c)$$

$$\le -\frac{\sigma}{N} \sum_{z_i \in D_-} \Phi_\theta(z_i) \le 0$$

$\square$

## B.3 PROOF OF LEMMA 2

**Lemma 2.** *When applying the relabeling function $\mathcal{R}$ in Eq. (12) over a training sample $z_i \in D$ with a class label $m$, the CE loss $l_i(\theta)$ at $z_i$ is changed from $-\log(\varphi_m(x_i, \theta))$ to $-\log(1 - \varphi_m(x_i, \theta))$.*

*Proof.* Recall that the model's prediction at $x_i$ is $\varphi(x_i, \theta) = (\varphi_1, \varphi_2, ..., \varphi_K)$ and our relabeling function is:

$$y'_{i,k} = \begin{cases} 0, & \text{if } k = m \\ \log_{\varphi_k} \sqrt[K-1]{1 - \varphi_m}, & \text{otherwise} \end{cases}$$

Here we assume the training example $z_i$ belongs to class $m$ which means that $y_{im} = 1$ and the other components in the one-hot vector $y_i$ are 0. The prime CE loss is $-\log(\varphi_m(x_i, \theta))$. If we use our relabeling function to change the label of $x_i$, the loss at $z_i$ will be:

$$
\begin{aligned}
\tilde{l}(z_i, \theta) &= -\sum_{k \neq m} \log_{\varphi_k} \sqrt[K-1]{1 - \varphi_m} \times \log(\varphi_k) \\
&= -\sum_{k \neq m} \frac{\log(\sqrt[K-1]{1 - \varphi_m})}{\log(\varphi_k)} \times \log(\varphi_k) \\
&= -\sum_{k \neq m} \frac{\log(1 - \varphi_m)}{K - 1} \\
&= -\log(1 - \varphi_m)
\end{aligned}
$$

In this way, if we use relabeling function $\mathcal{R}$ to change the label of example $z_i$, the loss function will be changed from $-\log(\varphi_m(x_i, \theta))$ to $-\log(1 - \varphi_m(x_i, \theta))$. □

### B.4 PROOF OF THEOREM 2

**Theorem 2.** *In multi-class classification, let $\varphi_y(x_i, \hat{\theta})$ denote the probability that $z_i$ is classified as its truth class label by the model with the optimal parameters $\hat{\theta}$ on $D$, and $\sigma$ be the infimum of $\frac{\varphi_y(x_i, \hat{\theta})}{1 - \varphi_y(x_i, \hat{\theta})}$. Relabeling the samples in $D_- = \{z_i \in D \mid \Phi_\theta(z_i) > 0\}$ with $\mathcal{R}$ leads to a test risk lower than the one achieved by discarding or downweighting $D_-$. Formally, we have:*

$$
L(Q, \hat{\theta}_{\epsilon \mathcal{R}}) - L(Q, \hat{\theta}_\epsilon) \approx -\frac{\sigma}{N} \sum_{z_i \in D_-} \Phi_\theta(z_i) \leq 0.
$$

*Proof.* According to Lemma 2, Eq. (8)and Eq. (10), we can estimate the change of test loss at a test sample $z_j^c \in Q$ caused by relabeling as follows:

$$
\begin{aligned}
l_i(z_j^c, \hat{\theta}_{\epsilon_i \mathcal{R}_i}) - l_i(z_j^c, \hat{\theta}) &\approx \epsilon_i \times \eta_{\theta \mathcal{R}}(z_i, z_j^c) \\
&= -\frac{1}{N} \frac{1}{1 - \varphi_y(x_i, \hat{\theta})} \Phi_\theta(z_i, z_j^c)
\end{aligned}
$$

Further, we can derive the following:

$$
\begin{aligned}
L(Q, \hat{\theta}_{\epsilon \mathcal{R}}) - L(Q, \hat{\theta}_\epsilon) &= L(Q, \hat{\theta}_{\epsilon \mathcal{R}}) - L(Q, \hat{\theta}) + L(Q, \hat{\theta}) - L(Q, \hat{\theta}_\epsilon) \\
&= \sum_{z_i \in D_-} \sum_{j=1}^{M} l(z_j^c, \hat{\theta}_{\epsilon_i \mathcal{R}_i}) - l(z_j^c, \hat{\theta}) - (l(z_j^c, \hat{\theta}_{\epsilon_i}) - l(z_j^c, \hat{\theta})) \\
&\leq \sum_{z_i \in D_-} (\sum_{j=1}^{M} \frac{1}{N} \eta_{\theta \mathcal{R}}(z_i, z_j^c) + \frac{1}{N} \sum_{j=1}^{M} \Phi_\theta(z_i, z_j^c)) \\
&= \frac{1}{N} \sum_{z_i \in D_-} \sum_{j=1}^{M} (\frac{-1}{1 - \varphi_y(x_i, \hat{\theta})} + 1) \Phi_\theta(z_i, z_j^c) \\
&\leq -\frac{\sigma}{N} \sum_{z_i \in D_-} \Phi_\theta(z_i) \leq 0
\end{aligned}
$$

□

## C  EXPERIMENTAL SETTINGS

### C.1  THE STATISTICS OF THE DATASETS

Table 5 shows the statistics of the datasets. We perform extensive experiments on public datasets from different domains to verify the effectiveness and robustness of our approach RDIA. All the datasets could be found in https://www.csie.ntu.edu.tw/cjlin/libsvmtools/datasets/.

Table 5: The statistics of the datasets.

| Dataset | #samples | #features | #classes | #domain |
|---------|----------|-----------|----------|---------|
| Breast-cancer | 683 | 10 | 2 | Medical |
| Diabetes | 768 | 8 | 2 | Medical |
| News20 | 19,954 | 1,355,192 | 2 | Text |
| Adult | 32,561 | 123 | 2 | Society |
| Real-sim | 72,309 | 20,958 | 2 | Physics |
| Covtype | 581,012 | 54 | 2 | Life |
| Criteo1% | 456,674 | 1,000,000 | 2 | CTR |
| Avazu | 14,596,137 | 1,000,000 | 2 | CTR |
| MNIST | 70,000 | 784 | 2/10 | Image |
| CIFAR10 | 60,000 | 3,072 | 2/10 | Image |

## C.2  TR-VA-TE DIVISIONS

We follow the Tr-Va-Te (Training/Validation/Test set divisions) setting in Wang et al. (2020b) to measure the generalization ability of our approach RDIA. Specifically, the influence of each training instance is estimated with the validation set using the validation loss and the model's performance is tested by an additional out-of-sample test set which ensures we do not utilize any information of the test data.

When training logistic regression, we randomly pick up 30% samples from the training set as the validation set. For different influence-based approaches, the training/validation/test sets are kept the same for fair comparison. Both MNIST and CIFAR10 are 10-classes image classification datasets while logistic regression can only handle binary classification. On MNIST, we select the number 1 and 7 as positive and negative classes, respectively; On CIFAR10, we perform binary classification on cat and dog. For each image, we convert all the pixels into a flattened feature vector where each pixel is scaled by 1/255.

When training deep models, due to the high time complexity of estimating influence functions, we randomly exclude 100 samples (1%) from the test sets of MNIST and CIFAR10 as the respective validation sets, and the remaining data is used for testing.

## C.3  IMPLEMENTATION DETAILS

We used the Newton-CG algorithm Martens (2010) to calculate the influence functions for the logistic regression model and applied Stochastic estimation Agarwal et al. (2017) for two deep models with 1000 clean data in the validation set. For logistic regression model, we select the regularization term C = 0.1 for fair comparison. We adopt the Adam optimizer with the learning rate of 0.001 to train the LeNet on MNIST. After calculating the influence functions and relabeling the identified harmful training samples using $\mathcal{R}$, we reduce the learning rate to $10^{-5}$ and update the models until convergence. For CIFAR10, we use the SGD optimizer with the learning rate of 0.01 and the momentum of 0.9 to train the CNN. Then we change the learning rate to 0.001 and update the models based on the relabeled training set. Here we use different optimizers to train the models. This indicates RDIA is independent of the update strategy used for model training. The batch size is set to 64 in all the experiments and the hyperparameter $\alpha$ is tuned with the validation set for best performance.

## D  EXTENSIVE ANALYSIS OF RDIA

### D.1  COMPLEXITY ANALYSIS

According to Koh & Liang (2017), the time complexity of calculating influence function for one training sample(i.e., Eq. 2) is $O(NP)$, where $N$ and $P$ stand for the sizes of training set and model's parameter set, respectively. Note that the time complexity of relabeling one sample is $O(N)$. Considering the complexity of calculating influence functions, the time cost of relabeling harmful samples is negligible which means our RDIA is as fast as any influence-based approaches.

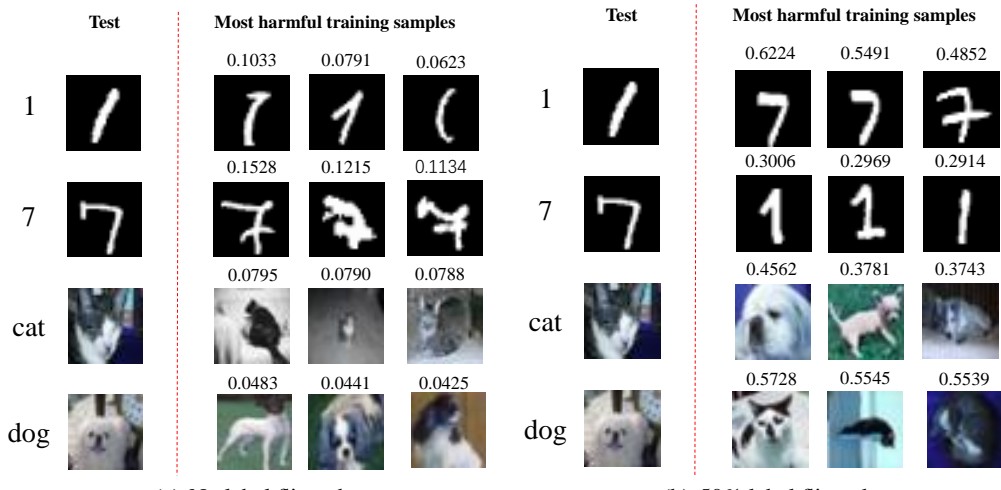

Figure 3: Identified harmful examples from MNIST and CIFAR10. For each test example, three harmful training samples with the highest influence estimates (above the images) are provided.

### D.2 RELATIONSHIP WITH PROPENSITY SCORE

Propensity score (Rosenbaum & Rubin, 1983; Bickel et al., 2009) is a well-studied technique to solve the distribution mismatch (also called covariate shift) problem where training and testing sets are sampled from two different distributions $P_{train}(x, y)$ and $P_{test}(x, y)$, respectively. Its basic idea is to assign the propensity score to each training sample to make the test risk unbiased. Unlike the influence function calculated by measuring the change of test loss, propensity score is calculated directly by estimating the probability of each training sample belonging to the test distribution. If we could estimate the training and test distribution accurately, we could also use the propensity score to replace the influence function for identifying whether the training sample is harmful. We leave it for the future work.

### E VISUALIZATION OF IDENTIFIED HARMFUL SAMPLES

We provide examples of harmful samples identified by influence functions to illustrate the effectiveness of influence analysis. We apply the logistic regression on MNIST (class 1 and 7) and CIFAR10 (class cat and dog). The influence functions are estimated by Newton-CG algorithm (Martens, 2010). We provide the three most harmful images which have the highest influence scores and share the same label with the test sample.

Figure 3(a) shows three identified harmful training images for each test image when there are no flipped labels in the training set. We can see that the identified harmful training samples are visually different from the original pictures, which disturbs the model's prediction on the test image. That is, the presence of clean but harmful training images would damage the model's performance.

Figure 3(b) shows the identified harmful training images when 50% labels of training data have been flipped. It is easy to see that the harmful images have corrupted labels, which confirms the effectiveness of applying influence analysis to locate noisy samples.

### F RDIA-LS: A LOSS-BASED RELABELING APPROACH

#### F.1 LIMITATIONS OF RDIA

In the main paper, we have discussed a novel data relabeling framework RDIA via influence analysis. Based on the advantages of influence functions, RDIA is able to handle different types of training biases and is agnostic to a specific model or data type. However, the time complexity of estimating

---

**Algorithm 2:** RDIA-LS

---

**Input:** Deep neural network $\theta$, learning rate $\beta$, trainig set $D$, training epoch $T$, iteration $N$, sample
   selection ratio $\rho$, underweight hyperparameter $\gamma$ such that $0 \leq \gamma \leq 1$.

1 **for** $t \in [1, \ldots, T]$ **do**
2      **Shuffle** training set $D$;
3      **for** $n \in [1, \ldots, N]$ **do**
4          **Fetch** $n$-th mini-batch $\bar{D}$ from $D$;
5          **Identify** harmful samples using training loss:                    **//Step I**
6          $\bar{D}_+ \leftarrow \arg\min_{\bar{D}:|\bar{D}| \geq \rho|\bar{D}|} L(\bar{D}, \theta)$;
7          $\bar{D}_- \leftarrow \bar{D} \setminus \bar{D}_+$;
8          **Relabel** the identified harmful training samples:            **//Step II**
9          $\bar{D}'_- \leftarrow \mathcal{R}(\bar{D}_-)$;
10         **Obtain** the loss as: $L_{\mathcal{R}} = \gamma L(\bar{D}'_-, \theta) + (1 - \gamma)L(\bar{D}_+, \theta)$
11         **Update** the model: $\theta \leftarrow \theta - \beta \nabla L_{\mathcal{R}}(\bar{D}_+, \theta)$;         **//Step III**
12      **end**
13 **end**

---

the influence of one training sample is $O(NP)$, where $N$ and $P$ stand for the sizes of training set and model's parameter set, respectively. This is relatively high for deep models which have thousands of parameters. Moreover, according to (Koh & Liang, 2017), the approximate estimation of influence functions on deep models may not be accurate and hence the second step of RDIA suffers from false positives and false negatives. When harmful samples account for the majority of the training set, e.g., high noise rates, it is difficult to filter most of harmful samples using the estimated influence.

## F.2 RDIA-LS

To address the aforementioned limitations, we aim to extend RDIA to solve the specific problem. Here we focus on combating noisy labels with deep models since label noise is usually a primary root cause of training bias. We notice that training loss has been used to filter out training samples with corrupted labels in many previous works (Arpit et al., 2017; Han et al., 2018; Wei et al., 2020; Yu et al., 2019). It is worth mentioning that the noisy training samples identified by training loss are not equivalent to the harmful ones identified by influence functions because the latter are evaluated to have negative influence on the test performance. Nevertheless, since the selected high-loss training samples are very likely to involve corrupted labels, applying our relabeling function over them has the potential of correcting corrupted labels and benefiting the test performance. Besides, using training loss to identify harmful samples is more efficient as it avoids the estimation of influence functions. Hence, we propose to use training loss to identify noisy samples and develop a loss-based data relabeling approach named RDIA-LS, which can be viewed as an extension of RDIA for combating corrupted labels with deep models.

RDIA-LS consists of three steps: *noisy samples identification*, *noisy samples relabeling* and *model updating*. It shares the same last two steps with RDIA. The only difference between RDIA-LS and RDIA is that RDIA-LS uses training loss to identify noisy samples in each training epoch so that it does not need to train the model until convergence first. Specifically, given a mini-batch of training instances $\bar{D} \subseteq D$, RDIA-LS feeds forward all the samples in $\bar{D}$ and then sorts them in an ascending order of their training losses. Following the prior works (He & Garcia, 2009), we regard the large-loss instances as noisy and the small-loss instances as clean. We use the rate of $\rho$ to select the possibly clean training instances in $\bar{D}$, i.e., $\bar{D}_+ = \arg\min_{\bar{D}:|\bar{D}| \geq \rho|\bar{D}|} L(\bar{D}, \theta)$. The remaining high-loss training instances are treated as noisy samples, i.e., $\bar{D}_- = \bar{D} \setminus \bar{D}_+$. We follow (Han et al., 2020) to determine the value of the selection ratio $\rho$. After we have $\bar{D}_-$, we use our relabeling function $\mathcal{R}$ to relabel the samples in $\bar{D}_-$ and then update the model with $\bar{D}_+ \cup \bar{D}'_-$. In our implementation, we simply modify the loss of the identified noisy samples based on Lemma 2 without performing actual relabeling. We use the hyperparameter $\gamma \in [0, 1]$ to control the model tendency of learning from the clean instances and the relabeled noisy instances. The detailed procedure of RDIA-LS is provided in Algorithm 2.

Table 6: Average test accuracy (±std) on MNIST over the last 10 epochs.

| Noise ratio ($\tau$) | 0.2 | 0.4 | 0.6 | 0.8 |
|---|---|---|---|---|
| ERM | 79.46 ± 0.42 | 59.12 ± 0.37 | 41.40 ± 0.05 | 23.43 ± 0.30 |
| S-model | 97.46 ± 0.15 | 83.52 ± 0.14 | 60.88 ± 0.32 | 41.63 ± 1.42 |
| F-correction | 98.02 ± 0.11 | 87.05 ± 0.05 | 74.15 ± 1.09 | 63.83 ± 1.76 |
| Self-teaching | 94.49 ± 0.13 | 92.49 ± 0.14 | 86.26 ± 0.27 | 75.95 ± 1.03 |
| Co-teaching | 97.89 ± 0.12 | 94.05 ± 0.07 | 90.72 ± 0.03 | 78.54 ± 0.21 |
| SIGUA | 97.94 ± 0.03 | 96.57 ± 0.02 | 93.84 ± 0.07 | 83.75 ± 0.15 |
| RDIA-LS | **98.12 ± 0.02** | **97.57 ± 0.05** | **95.32 ± 0.06** | **87.85 ± 0.21** |

Table 7: Average test accuracy (±std) on CIFAR10 over the last 10 epochs.

| Noise ratio ($\tau$) | 0.2 | 0.4 | 0.6 | 0.8 |
|---|---|---|---|---|
| ERM | 71.84 ± 1.07 | 55.62 ± 0.31 | 35.56 ± 0.22 | 16.90 ± 0.62 |
| S-model | 76.83 ± 0.72 | 65.37 ± 0.39 | 43.79 ± 0.15 | 17.41 ± 0.08 |
| F-correction | 80.91 ± 0.16 | 71.68 ± 0.65 | 57.51 ± 0.24 | 19.63 ± 0.78 |
| Self-teaching | 78.92 ± 0.21 | 70.91 ± 0.26 | 62.76 ± 0.05 | 20.32 ± 0.13 |
| Co-teaching | 79.43 ± 0.11 | 72.88 ± 0.08 | 66.23 ± 0.32 | 22.47 ± 0.15 |
| SIGUA | 81.58 ± 0.36 | 74.43 ± 0.11 | 66.28 ± 0.14 | 24.26 ± 0.23 |
| RDIA-LS | **82.94 ± 0.19** | **77.26 ± 0.14** | **67.52 ± 0.21** | **25.35 ± 0.17** |

We conduct the additional experiments in Appendix G to empirically show that RDIA-LS is effective and efficient to handle training data with corrupted labels for deep learning, which spotlights the great scalability of our approach RDIA.

# G    PERFORMANCE EVALUATION OF RDIA-LS

We now conduct the experiments to evaluate the effectiveness and efficiency of RDIA-LS using DNNs on MNIST, CIFAR10, CIFAR100 and Clothing1M. The first three datasets are clean and corrupted artificially. Clothing1M is a widely used real-world dataset with noisy labels (Patrini et al., 2017).

## G.1    IMPLEMENTATION DETAILS

We apply the same network structures used in the main paper and use LeNet (2 convolutional layers and 1 fully connected layer) for MNIST, a CNN with 6 convolutional layers followed by 2 fully connected layers used in (Wang et al., 2019) for CIFAR10 and CIFAR100, and a 18-layer ResNet for Clothing1M. We follow the settings in (Han et al., 2018) for all the comparison methods. Specifically, for MNIST, CIFAR10 and CIFAR100, we use the Adam optimizer with the momentum of 0.9, initial learning rate of 0.001, and the batch size of 128. We run 200 epochs (T=200) in total and linearly decay the learning rate till zero from 80 to 200 epochs. As for Clothing1M, we use the Adam optimizer with the momentum 0.9 and set the batch size to be 64. We run 15 epochs in total and set learning rate to $8 \times 10^{-4}$, $5 \times 10^{-4}$ and $5 \times 10^{-5}$ for average five epochs.

We set the ratio of small-loss instances as $\rho = 1 - \min\{\frac{t}{T_k} * \tau, \tau\}$ which is changed dynamically with the current training epoch $t$ and $T_k = 5$ for Clothing1M and $T_k = 10$ for the other datasets. In this way, we can determine $D_-$ and $D_+$ in each training epoch. If the noise ratio $\tau$ is not known in advance, we could use the method (Yu et al., 2018) to estimate $\tau$. The hyperparameter $\gamma$ is tuned in $\{0.05, 0.10, 0.15, \cdots, 0.95\}$ with the validation set for best performance. If there is no validation set, we could use training loss to select a clean subset from the training set as the validation set. Following loss-based approach (Han et al., 2020; 2018; Jiang et al., 2018), we use the test accuracy as the metric, i.e., (#correct predictions) / (#test instances).

Table 8: Average test accuracy (±std) on CIFAR100 over the last 10 epochs.

| Noise ratio ($\tau$) | 0.2 | 0.4 | 0.6 | 0.8 |
|---|---|---|---|---|
| ERM | 35.14 ± 0.44 | 20.58 ± 0.23 | 12.87 ± 0.42 | 4.41 ± 0.14 |
| S-model | 45.71 ± 0.15 | 34.94 ± 1.29 | 19.82 ± 0.67 | 2.61 ± 1.18 |
| F-correction | 47.51 ± 0.24 | 37.91 ± 1.47 | 22.75 ± 1.87 | 2.10 ± 2.23 |
| Self-teaching | 47.37 ± 0.30 | 40.55 ± 0.04 | 30.62 ± 0.24 | 13.49 ± 0.37 |
| Co-teaching | 47.15 ± 0.16 | 41.41 ± 0.62 | 30.78 ± 0.11 | 15.15 ± 0.46 |
| SIGUA | 48.52 ± 0.21 | 42.93 ± 0.15 | 30.73 ± 0.41 | 14.31 ± 0.02 |
| RDIA-LS | **50.24 ± 0.15** | **44.20 ± 0.11** | **32.67 ± 0.17** | **20.21 ± 0.04** |

Table 9: Average test accuracy (±std) results on Clothing1M.

| Methods | ERM | F-correction | Co-teaching | SIGUA | RDIA-LS |
|---|---|---|---|---|---|
| Accuracy(%) | 64.54 ± 1.05 | 69.13 ± 0.25 | 68.36 ± 0.35 | 69.35 ± 0.41 | **69.64 ± 0.14** |

## G.2 COMPARISON METHODS

We compare our proposed **RDIA-LS** with the following baselines. S-model (Goldberger & Ben-Reuven, 2017) and F-correction (Patrini et al., 2017) are the existing data relabeling approach which aims to estimate the noisy transition matrix to correct the noisy labels. The last three approaches are the state-of-the-art loss-based resampling approaches. (1) **ERM**: it trains one network with all the training data using cross-entropy loss. (2) **S-model** (Goldberger & Ben-Reuven, 2017): it uses an additional softmax layer to model the noise transition matrix to correct the model (3) **F-correction** (Patrini et al., 2017): it corrects the prediction by the noise transition matrix estimated by the other network. (4) **Self-teaching** (Jiang et al., 2018): it trains one network with only the selected small-loss instances $D_+$. (5) **Co-teaching** (Han et al., 2018): it trains two networks simultaneously and improves self-teaching by updating the parameters of each network with the small-loss instances $D_+$ selected by the peer network. (6) **SIGUA** (Han et al., 2020): it trains one network with the selected small-loss instances $D_+$ and high-loss instances $D_-$ via gradient descent and gradient ascent, respectively.

## G.3 EXPERIMENTAL RESULTS

**Comparison with the Baselines.**

RDIA-LS is proposed to combat noisy labels for deep learning. In order to evaluate how RDIA-LS improves the robustness of deep models, we perform experiments on MNIST+LeNet, CIFAR10+CNN and CIFAR100+CNN with different noise ratios and the real-world noisy dataset Clothing1M+Resnet18. The average results of test accuracy are reported in Table 6, Table 7, Table 8 and Table 9 . We have the following observations. (1) RDIA-LS achieves the highest test accuracy in all the cases. When noise ratio is 0.2, the improvement of RDIA-LS is relatively small. This is reasonable as the performance gain of RDIA-LS obtained from utilizing noisy data is restricted due to the low noise ratio. When the noise ratio exceeds 0.4, RDIA-LS significantly outperforms the existing loss-based approaches, achieving up to 5% relative improvement in test accuracy. It indicates that RDIA-LS can still effectively reuse harmful training instances to improve model's robustness under high noise ratios. (2) RDIA-LS consistently performs better than S-model, F-correction, and SIGUA, which implies that using $\mathcal{R}$ to relabel noisy training samples identified by training loss is more effective than modeling the noise transition matrix or performing gradient ascent with identified noisy training instances. (3) RDIA-LS outperforms all the baselines on the real-world noisy dataset Clothing1M, which demonstrates the effectiveness of applying RDIA-LS in practice.

**Comparison with RDIA.**

Table 10 reports the running time of harmful/noisy samples identification in RDIA and RDIA-LS. We exclude the results of LS on logistic regression since training loss can only be used to filter out noisy samples for training deep models. From the table, we can see that using influence function to identify harmful samples for logistic regression is efficient. However, when training deep models with millions of parameters, using training loss to filter out noisy samples is much more efficient.

Table 10: Time cost of identifying harmful samples.

| Dataset | Model | RDIA | RDIA-LS |
|---------|-------|------|---------|
| Diabetes | LR | 0.03 sec | - |
| News20 | LR | 1.8 sec | - |
| Criteo1% | LR | 7.1 sec | - |
| Avazu | LR | 4.2 min | - |
| MNIST | LeNet | 4~5 hour | 0.1 sec |
| CIFAR10 | CNN | 7~9 hour | 0.6 sec |

RDIA-LS is an extension of RDIA to combat noisy samples with deep models. The aforementioned experimental results show that RDIA-LS is effective and efficient to handle training data with corrupted labels for deep learning. However, it is worth noticing that RDIA-LS relies on the small-loss trick that the samples which have the larger training loss may contain the corrupted labels. In this way, RDIA-LS is only suitable for the deep models against the corrupted labels and could fail in the situation where the small-loss trick does not hold while RDIA has no such constraint.

## H  ADDITIONAL DISCUSSION ON DATA RELABELING

Existing relabeling approaches (Goldberger & Ben-Reuven, 2017; Jiang et al., 2018; Lee et al., 2018) are proposed to combat noisy labels with DNNs. They focus on estimating the noise transition matrix to convert the corrupted labels to clean labels. However, current relabeling methods suffer from two limitations. First, they aim to find the true labels of the training samples, which means they can only deal with label noise. Second, they employ some additional structures to correct the labels, which are dependent of the specific model structures. For example, Goldberger et al. (Goldberger & Ben-Reuven, 2017) added an additional softmax layer to present the noise transition matrix and CleanNet (Lee et al., 2018) used the auto-encoder to update the corrupted labels. In contrast, we aim to develop a relabeling function based on influence analysis to change the labels of harmful samples towards better model performance. We do not require output labels to be one-hot vector since our objective is not to find the truth labels of training samples. Besides, we extend our approach to RDIA-LS to effectively combat noisy samples for training DNNs, which outperforms the existing data relabeling approaches (Goldberger & Ben-Reuven, 2017; Jiang et al., 2018).

DUTI (Zhang et al., 2018) is an effective data relabeling approach which could debug and correct the wrong labels in the training set. It uses a bi-level optimization scheme to recommend the most influential training samples for cleaning and suggest the possibly cleaned labels. The proposed relabeling function in DUTI is different from our approach. Specifically, the relabeling function in DUTI is trained by a bi-level optimization using the gradient of the validation loss while our proposed relabeling function has nothing to do with gradient, validation loss.

