# OpenReview forum: "Resolving Training Biases via Influence-based Data Relabeling"
_ICLR.cc/2022/Conference — ICLR 2022 Oral_

### Official Review · Reviewer_FfyB · 2021-10-27

**Correctness:** 4
**Technical Novelty And Significance:** 2
**Empirical Novelty And Significance:** 3
**Recommendation:** 6
**Confidence:** 4

**Main Review:**

## Strength
The authors empirically demonstrated that the label correction is much more effective than the standard sampling/data reweighting-based approach for model improvement.
This result is coherent with the one previously reported in [Ref1] where a very similar data relabeling approach was studied.
The results on DNNs (on MNIST and CIFAR10) are new where [Ref1] considered only kernel-based models.

* [Ref1] Training Set Debugging Using Trusted Items, AAAI 2018.
    * *I am not the author of [Ref1].*

## Weakness
The proposed method is very similar to the one proposed in [Ref1], while the current paper misses this important prior work.
[Ref1] formulated the data relabeling problem as the bi-level optimization, and proposed an algorithm to optimize the amount of relabeling using gradient descent.
An important point here is that the gradient of the amount of relabeling considered in [Ref1] is essentially identical to the relabeling criteria proposed in (10) in this paper (see below for the detail).
In addition, once we interpret the proposed label correction criteria as the gradient descent update, Lemma1 and Theorem1 confirming the decrease of the validation loss are more or less trivial (unless the amount of correction is too large).
In summary, although the ways the amount of relabeling is optimized are different (using gradient descent or by one step update), I believe the close connection between [Ref1] and the proposed method needs to be described appropriately in the paper, e.g., through detailed discussions and experimental comparisons.

*Close connection to [Ref1]*

As the special case of the formulation of [Ref1], we can obtain the following bi-level optimization (here, I removed some terms from the original formulation in [Ref1] for simplicity, and used the notation of the current paper).

$$
\min\_{\delta \in \mathbb{R}^{n}} L(Q; \hat{\theta}\_\delta) := \frac{1}{M} \sum\_{j=1}^{M} l\_j^c(\hat{\theta}\_\delta) , \\: \text { s.t. } \\: \hat{\theta}\_\delta = \underset{\theta \in \mathbb{R}^{p}}{\operatorname{argmin}} \frac{1}{N} \sum\_{i=1}^{N} l\_i(z\_{i\delta\_i}, \theta) ,
$$

where $\delta$ is the amount of relabeling.
This is the optimization of the amount of relabeling so that the the validation loss to be minimized.
Following [Ref1], we can derive the gradient of the validation loss $L(Q; \hat{\theta}_\delta)$ with respect to $\delta_i$ as

$$
\frac{dL(Q; \hat{\theta}\_\delta)}{d\delta\_i} = \nabla\_\theta L(Q; \hat{\theta}\_\delta) \frac{d\hat{\theta}\_\delta}{d\delta\_i} = -\frac{1}{N}\nabla\_\theta L(Q; \hat{\theta}\_\delta)^\top H\_{\hat{\theta}\_\delta}^{-1} \frac{\partial l\_i(z_{i\delta_i}, \hat{\theta}\_\delta)}{\partial \delta\_i} .
$$

When there is no label correction, $\delta = 0$ and we have $\hat{\theta}_0$.

Then, if we consider relabeling the $i$-th training instance with the gradient step size $\epsilon$, we have the update

$$
\delta\_i \leftarrow 0 - \epsilon \left. \frac{dL(Q; \hat{\theta}\_\delta)}{d\delta\_i} \right|\_{\delta\_i = 0}
$$

and the decrease of the validation loss induced by this update is

$$
L(Q; \hat{\theta}\_{\delta}) - L(Q; \hat{\theta}\_0) \approx \epsilon \left. \frac{dL(Q; \hat{\theta}\_\delta)}{d\delta\_i} \right|\_{\delta\_i = 0}
$$
$$
= \frac{\epsilon}{N} \nabla\_\theta L(Q; \hat{\theta}\_0)^\top H\_{\hat{\theta}\_0}^{-1} \frac{\partial l\_i(z\_{i\delta\_i}, \theta_0)}{\partial \delta\_i}
$$
$$
\approx \frac{1}{N}\nabla\_\theta L(Q; \hat{\theta}\_0)^\top H\_{\hat{\theta}\_0}^{-1} (l\_i(z\_{i \epsilon}, \theta) - l\_i(\theta)) .
$$
The last line is nothing but the criteria (10) proposed in this paper.

---
## After Discussion with Authors

I conducted an experiment on breast cancer dataset by myself (see below).
There, I confirmed that RDIA does better than the One-Step GD update of [Ref1].
I therefore decided to increase my score with a strong expectation to the authors for **referring [Ref1] appropriately in the main body of the paper**.
As I mentioned in my original review, this is not the first study considering relabeling based on the influence function-like technique.
Recalling that influence function is one specific example of implicit gradient, [Ref1] is the first work in this direction to my knowledge (even if [Ref1] does not describe the fact explicitly).
As I demonstrated in my code, the simplest version of [Ref1] with only one-step update (and without any human intervention) does good job.
**I strongly expect the authors to pay certain respect to [Ref1] and do not underestimate their technical contribution**.
For example, the current writing in Appendix H seems to be not appropriate.
Even without human intervention, DUTI [Ref1] can do the same (i.e., relabel the training bias towards better model performance) as the proposed approach (but with slightly inferior performance).
> (2) The learning task of DUTI is to debug the training instances which may contain the wrong labels and predict the true labels. While the target of our approach is to relabel the training bias towards better model performance. In this way, only our approach could relabel the biased training samples with correct labels towards better model performance.

### The results on breast cancer dataset

*Test Accuracy*

|     |   ERM |   One-Step GD |   RDIA |
|----:|------:|--------------:|-------:|
| 0   | 0.959 |         0.96  |  0.961 |
| 0.1 | 0.96  |         0.968 |  0.962 |
| 0.2 | 0.953 |         0.951 |  0.959 |
| 0.3 | 0.931 |         0.944 |  0.957 |
| 0.4 | 0.842 |         0.897 |  0.958 |
| 0.5 | 0.563 |         0.762 |  0.958 |
| 0.6 | 0.178 |         0.782 |  0.953 |
| 0.7 | 0.075 |         0.875 |  0.953 |
| 0.8 | 0.05  |         0.931 |  0.949 |
| 0.9 | 0.037 |         0.95  |  0.94  |

*Test Loss*

|     |      ERM |   One-Step GD |     RDIA |
|----:|---------:|--------------:|---------:|
| 0   | 0.141025 |      0.161033 | 0.13819  |
| 0.1 | 0.237151 |      0.24697  | 0.146115 |
| 0.2 | 0.345921 |      0.347718 | 0.157343 |
| 0.3 | 0.453018 |      0.451272 | 0.171787 |
| 0.4 | 0.56437  |      0.557865 | 0.185989 |
| 0.5 | 0.698697 |      0.63033  | 0.197167 |
| 0.6 | 0.876071 |      0.590184 | 0.203124 |
| 0.7 | 1.08732  |      0.492453 | 0.196512 |
| 0.8 | 1.44151  |      0.362804 | 0.197389 |
| 0.9 | 2.0108   |      0.265325 | 0.241273 |

```
import numpy as np
import pandas as pd
from sklearn import datasets
from sklearn.model_selection import train_test_split
from sklearn.preprocessing import StandardScaler
from IPython.display import display

def sigmoid(u):
    return 1.0 / (1.0 + np.exp(-u))

def logloss(w, b, x, y):
    p = sigmoid(x.dot(w) + b)
    return - (y.dot(np.log(p)) + (1 - y).dot(np.log(1 - p))) / y.size

def logreg(x, y, lam=0.1, lr=0.01, max_itr=100):
    num, dim = x.shape
    xo = np.concatenate([x, np.ones((num, 1))], axis=1)
    w = np.zeros(dim+1)
    for itr in range(max_itr):
        p = sigmoid(xo.dot(w))
        g = (p - y).dot(xo) / num
        g[:-1] = g[:-1] + lam * w[:-1]
        w = w - lr * g
    return w[:-1], w[-1]

# experiment
lam = 1e-2
acc, loss = [], []
nr = np.linspace(0, 0.9, 10)
for noise_rate in nr:
    acc_n, loss_n = [], []
    for seed in range(10):

        # breast cancer data
        # train / val / test = 300 / 169 / 100
        x, y = datasets.load_breast_cancer(return_X_y=True, as_frame=False)
        x, xte, y, yte = train_test_split(x, y, test_size=269, random_state=seed)
        xval, xte, yval, yte = train_test_split(xte, yte, test_size=100, random_state=seed)

        # normalize features
        scaler = StandardScaler().fit(x)
        x = scaler.transform(x)
        xval = scaler.transform(xval)
        xte = scaler.transform(xte)

        # noisy label in training
        np.random.seed(seed)
        flip = np.random.rand(y.size) < noise_rate
        y = np.logical_xor(y, flip).astype(int)

        # fit logreg
        w, b = logreg(x, y, max_itr=500, lam=lam)

        # test accuracy / loss
        zte = (xte.dot(w) + b > 0).astype(int)
        acc_te1 = np.mean(yte == zte)
        loss_te1 = logloss(w, b, xte, yte)

        # influence function
        num, dim = x.shape
        xo = np.concatenate([x, np.ones((num, 1))], axis=1)
        p = sigmoid(x.dot(w) + b)
        H = xo.T.dot((p * (1 - p))[:, np.newaxis] * xo) / num
        H = 0.5 * (H + H.T) + np.diag([lam] * dim + [0]) # Hessian
        g = ((p - y)[:, np.newaxis] * xo) / num # gradient
        f = - np.linalg.solve(H, g.T).T # influence function

        # label gradient over validation loss
        xv = np.concatenate([xval, np.ones((xval.shape[0], 1))], axis=1)
        q = sigmoid(xval.dot(w) + b)
        gv = (q - yval).dot(xv) / xval.shape[0] # gradient of validation loss
        gy = f.dot(gv) * (np.log(p) - np.log(1 - p)) # label gradient

        # label correction by one-step gradient descent
        ynew = y - 1e+3 * gy # step size of GD = 1e+3
        ynew = np.minimum(1, np.maximum(0, ynew)) # clip label
        wnew, bnew = logreg(x, ynew, max_itr=500, lam=lam)

        # test accuracy / loss after label correction
        zte = (xte.dot(wnew) + bnew > 0).astype(int)
        acc_te2 = np.mean(yte == zte)
        loss_te2 = logloss(wnew, bnew, xte, yte)

        # label correction by RDIA
        ynew = y.copy()
        #ynew[f.dot(gv) > 0] = 1 - ynew[f.dot(gv) > 0] # flip label
        ynew[f.dot(gv) > 1e-3] = 1 - ynew[f.dot(gv) > 1e-3] # flip label
        wnew, bnew = logreg(x, ynew, max_itr=500, lam=lam)

        # test accuracy / loss after label correction
        zte = (xte.dot(wnew) + bnew > 0).astype(int)
        acc_te3 = np.mean(yte == zte)
        loss_te3 = logloss(wnew, bnew, xte, yte)

        acc_n.append((acc_te1, acc_te2, acc_te3))
        loss_n.append((loss_te1, loss_te2, loss_te3))

    acc.append(acc_n)
    loss.append(loss_n)
acc = np.array(acc)
loss = np.array(loss)

acc = pd.DataFrame(np.mean(acc, axis=1))
acc.columns = ['ERM', 'One-Step GD', 'RDIA']
acc.index = nr
loss = pd.DataFrame(np.mean(loss, axis=1))
loss.columns = ['ERM', 'One-Step GD', 'RDIA']
loss.index = nr
print('Test Accuracy')
display(acc)
print('Test Loss')
display(loss)
```

**Summary Of The Paper:**

This paper proposes a data relabeling technique using influence function.
The authors first derived the influence function for data relabeling as follows.

$$
\eta_{\theta \delta}\left(z\_i, z\_j^{c}\right) \left. \triangleq \frac{d l\_j^c \left(\hat{\theta}\_{\epsilon\_i \delta\_j}\right)}{d \epsilon\_i}\right|_{c\_i=0} =-\nabla\_{\theta} l\_j^c(\hat{\theta}) H\_{\hat{\theta}}^{-1}\left(\nabla\_\theta l\_i\left(z\_{i \delta}, \hat{\theta}\right)-\nabla\_\theta l\_i(\hat{\theta})\right)
$$

The authors then derived the specific expression for the cross-entropy loss:
$$
\eta_{\theta R}\left(z_{i}, z_{j}^{c}\right)=-\nabla_{\theta} l_{j}(\hat{\theta}) H_{\hat{\theta}}^{-1} w\left(z_{i}, \hat{\theta}\right)=-\nabla_{\theta} l_{j}(\hat{\theta}) H_{\hat{\theta}}^{-1}\left(-\frac{\nabla_{\theta} l_{i}(\hat{\theta})}{1-\varphi\left(x_{i}, \hat{\theta}\right)}\right)=\frac{-\Phi_{\theta}\left(z_{i}, z_{j}^{c}\right)}{1-\varphi\left(x_{i}, \hat{\theta}\right)}
$$
The authors experimentally demonstrated that data relabeling is more effective than the standard sampling/data reweighting-based approach for model improvement.

**Summary Of The Review:**

The proposed method is very similar to the one proposed in [Ref1], while the current paper misses this important prior work.
Although the ways the amount of relabeling is optimized are different (using gradient descent or by one step update), I believe the close connection between [Ref1] and the proposed method needs to be described appropriately in the paper, e.g., through detailed discussions and experimental comparisons.

* [Ref1] Training Set Debugging Using Trusted Items, AAAI 2018.

The strong aspect of this paper over [Ref1] is the experimental results where the results on DNNs (on MNIST and CIFAR10) are reported, while [Ref1] considered only kernel-based models.

---
## After Discussion with Authors

I conducted an experiment on breast cancer dataset by myself (see below).
There, I confirmed that RDIA does better than the One-Step GD update of [Ref1].
I therefore decided to increase my score with a strong expectation to the authors for **referring [Ref1] appropriately in the main body of the paper**.
As I mentioned in my original review, this is not the first study considering relabeling based on the influence function-like technique.
Recalling that influence function is one specific example of implicit gradient, [Ref1] is the first work in this direction to my knowledge (even if [Ref1] does not describe the fact explicitly).
As I demonstrated in my code, the simplest version of [Ref1] with only one-step update (and without any human intervention) does good job.
**I strongly expect the authors to pay certain respect to [Ref1] and do not underestimate their technical contribution**.

---

> ### Author Response · Authors · 2021-11-12
> **Response to Reviewer FfyB (1/3)**
>
> Thank you for your detailed feedback. We would like to clarify in detail the main concern about the similarity between our approach and [1].
>
>  >The proposed method is very similar to the one proposed in [1]. Although the ways the amount of relabeling is optimized are different (using gradient descent or by one step update), I believe the close connection between [Ref1] and the proposed method needs to be described appropriately in the paper, e.g., through detailed discussions and experimental comparisons.
>
> Indeed, we respectfully disagree with the reviewer and would like to explain that our method is **quite different** from [1] in the following aspects:
>
>
>
> - **Target:** The target of [1] is to debug the training instance with the wrong label and predict the true label, while the target of our approach is to relabel the training bias (may have the correct label) towards better model performance. It’s worth noticing that: (1) we do not require the relabeled y belongs to any explicit class, but require the relabeled y to reduce the validation loss; (2) our method would also relabel the biased training samples that have correct labels if such relabeling improves model performance while [1] can only correct the wrong labels. Since the targets of our approach and [1] are different, their solution is unsuitable to address our problem of resolving training biases. Specifically, [1] uses the budget b (predefined by the domain expert) to control the amount of detected wrong labels in the training set and then asks experts to check whether the data in b contains wrong labels and whether the method predicts the correct label. However, when applying [1] to relabel the training bias (even if there are some wrong labels), we have no idea how much data should be relabeled in the training set so that a lower validation loss can be achieved. **This means [1] is only suitable for debugging wrong labels in the training set, but hard to relabel the training bias for improving model performance** (RDIA works since we use the influence function to identify the harmful training samples and then relabel them).
>
> - **Relabeling Function and Optimization:** The most important thing is that the proposed relabeling function and optimization in [1] are totally different from our approach. In [1], a soft label is learned to correct the wrong label. **The relabeling function  $\delta$  in [1] is trained by a bi-level optimization using the gradient of the validation loss** which means the validation set is used for model training and the relabeling function is determined with the validation loss. However, we would like to clarify that our approach only uses the validation set to calculate the influence function without training on the validation set. More importantly, as shown in Equation (12) of our paper, **we propose an explicit relabeling function which has nothing to do with gradient, validation loss or even influence function.** Our method is close to prior influence-based methods [2,3] where the proposed reweighting/relabeling function is not obtained through optimization and will not be updated.
>
> - **Theoretical Guarantee:** Note that not every relabeling function can reduce validation loss theoretically. One of the main contributions of our paper is to theoretically prove that using our proposed relabeling function could achieve lower validation loss than resampling or reweighting approaches and this **gives a theoretical guarantee on the effectiveness of our approach**. However, we have found that [1] has no theoretical guarantee for its own method, making it even difficult to compare with the existing resampling approaches in theory. We regret that the reviewer has ignored our theoretical contribution.

---

> > ### Author Response · Authors · 2021-11-12
> > **Response to Reviewer FfyB (2/3)**
> >
> > - **Experiments:** The experiments in [1] report the precision-recall (PR) curve with respect to true bugs (the identified samples actually have the wrong labels) in the training set while our experiments report the predictive performance in the test set. Since the experimental settings and metrics are totally different in two works, we could not conduct the comparison experiments. Specifically,  as **our approach does not focus on finding true labels, it does not make sense for our approach to measure the precision of label correction as [1]**. Besides, when we apply [1] to relabel the training bias, there are two main problems. (1) [1] can only correct corrupted labels but cannot handle the clean but biased training samples. In contrast, our proposed relabeling approach could relabel the biased training samples that may have correct labels towards better model performance. (2) We would like to clarify that indeed there is no criteria in [1] to indicate which data should be relabeled. If we simply use all the predicted labels to replace the original labels, it would add new noises when the predicted labels are incorrect. As shown in Figure 3 of [1], the precision-recall (PR) curve shows that when the flagged bugs (i.e., the label predicted by [1] is different from its original label) contain 50% true noisy data, the precision of label correction is also about 50%. This indicates that, without expert knowledge, it is difficult to relabel noisy data only without changing the original labels of clean data, **making it unsuitable to apply [1] to our problem, e.g., resolving training biases towards better model performance.**
> >
> > > An important point here is that the gradient of the amount of relabeling considered in [1] is essentially identical to the relabeling criteria proposed in (10) in this paper. Once we interpret the proposed label correction criteria as the gradient descent update, Lemma1 and Theorem1 confirming the decrease of the validation loss are more or less trivial (unless the amount of correction is too large)
> >
> > For the convenience of discussion, we briefly copy the derivation given by the reviewer here:
> > If we set $\delta_i$ as a trainable parameter and optimize it in each training epoch with the single gradient descent step size $\epsilon$ , i.e., $\delta_i = 0-\epsilon\frac{\mathrm{d}L(Q;\hat{\theta}_\delta)}{\mathrm{d}\delta_i}$. Then we derive that
> > $$
> > L(Q;\hat{\theta}_\delta)-L(Q;\hat{\theta}_0) \approx \frac{1}{N} \nabla_\theta  L(Q;\hat{\theta})^T H_\hat{\theta_0}^{-1} \frac{\partial l_i(z_i\delta_i,\theta_0)}{\partial \delta_i}
> > $$
> >
> > $\bullet$  **First, we would like to clarify that our equation (10) is not identical to the above formula.**
> >
> > (1) The most important thing is that the our proposed relabeling function $\delta$ is not a trainable parameter **which means we could not derive** $\frac{\partial l_i(z_i\delta_i,\theta_0)}{\partial \delta_i}$ **since there is no gradient on**  **$\delta$**  **.** This also verifies our method is fundamentally different from [1] in relabeling function and optimization.
> >
> > (2) Indeed, we understand that reviewer concerned about the similarity between our method and [1] in measuring the change of validation loss, i.e., $L(Q;\hat{\theta}_\delta)-L(Q;\hat{\theta}_0)$. However, our equation is different from the above equation. As we mentioned in section 2, we follow [3,4] to derive the validation loss change as (Equation (10)):
> >
> > $$
> > L(Q;\hat{\theta}_\delta)-L(Q;\hat{\theta}_0) \approx \frac{1}{N} \nabla_\theta  L(Q;\hat{\theta}_0)^T H_\hat{\theta_0}^{-1} \nabla_\theta (l_i(z_i\delta,\hat{\theta}_0)-l_i(\hat{\theta}))
> > $$
> >
> > **Here we can clearly see the difference between the two equations, i.e.,**
> > $$
> > \nabla_\theta (l_i(z_{i\delta},\hat{\theta}_0)-l_i(\hat{\theta})) \neq \frac{\partial l_i(z_i\delta_i,\theta_0)}{\partial \delta_i}
> > $$
> > The former is the derivation of the model parameters and the latter is the derivation of the relabeling parameters.
> >
> > (3) Maybe it is our fault, but we do not understand how to get the last step of your derivation, i.e.,
> >
> > $$
> > \frac{1}{N} \nabla_\theta L(Q;\hat{\theta}_0)^TH_\hat{\theta_0}^{-1} \frac{\partial l_i(z_i\delta_i,\theta_0)}{\partial \delta_i}  \approx \frac{1}{N}\nabla_\theta L(Q;\hat{\theta}_0)^TH_\hat{\theta_0}^{-1}(l_i(z_i\delta,\hat{\theta}_0)-l_i(\hat{\theta}))
> > $$
> >
> > It seems like  $\frac{\partial l_i(z_i\delta_i,\theta_0)}{\partial \delta_i}  \approx  l_i(z_{i\delta},\hat{\theta}_0)-l_i(\hat{\theta})$ . However, $\delta$  here is a trainable parameter and it cannot be omitted. Please kindly let us know what we misunderstood.

---

> > > ### Author Response · Authors · 2021-11-12
> > > **Response to Reviewer FfyB (3/3)**
> > >
> > > $\bullet$ **Second, we would like to clarify that Lemma1 and Theorem1 confirming the decrease in the validation loss are non-trivial.**
> > >
> > > Recall that the validation loss decreases in [1] because it sets $\delta_i = 0-\epsilon\frac{\mathrm{d}L (Q;\hat{\theta}_\delta)}{\mathrm{d}\delta_i}$. In their approach, as we directly set $\delta$ to be the gradient of minimizing the validation loss in each training epoch, the decrease in the validation loss is somehow a trivial conclusion.
> > >
> > > However, like the existing influence-based resampling/reweighting approaches [2,3,5], our proposed relabeling function $\delta$  is not a trainable parameter which means we could not set $\delta$ according to the gradient of the validation loss (actually, we do not train the model with the validation set to update any parameters). To this end, how to develop a relabeling function without trainable parameters that has a theoretical guarantee on improving the model’s performance is a non-trivial task. It is even more challenging to develop a relabeling approach that is theoretically better than the existing influence-based resampling approaches, which is what we did in this paper.
> > >
> > >
> > >
> > > **To sum up, we believe that our approach is quite different from [1] in various aspects. We thank you again for your valuable comments and we are happy to revise the paper and add [1] as related work and discuss the main differences in Appendix H, which we believe would make our paper more comprehensive.**
> > >
> > >
> > >
> > > We hope we have addressed your questions; please let us know if there are any additional concerns.
> > >
> > >
> > >
> > > Reference
> > >
> > > [1] Training Set Debugging Using Trusted Items, AAAI 2018.
> > >
> > > [2] Data dropout: Optimizing training data for convolutional neural networks, ICTAI 2018.
> > >
> > > [3] Less Is Better: Unweighted Data Subsampling via Influence Function, AAAI2020.
> > >
> > > [4] Understanding Black-box Predictions via Influence Functions. ICML 2017.
> > >
> > > [5] Optimal Subsampling with Influence Functions. NIPS 2018.

---

> > > > ### Comment · Reviewer_FfyB · 2021-11-17
> > > > **Reply (1/3)**
> > > >
> > > > I would like to thank the authors for the detailed feedback.
> > > >
> > > > ## Target
> > > > Let me rephrase the feedbacks first.
> > > > The current paper and [1] are different in a sense that
> > > >
> > > > *1. The Objective of Relabeling*
> > > > * [1] is interested in detecting wrong labels and correct them.
> > > > * The current study focuses on improving validation loss.
> > > >
> > > > *2. Target of Relabeling*
> > > > * [1] limits some of the instances (trusted items) to keep the original labels, and optimize the labels of the remaining instances (untrusted items) only.
> > > > * The current study considers relabeling possibly all the instances.
> > > >
> > > > *3. The Output of Relabeling*
> > > > * [1] outputs labels in the output domain, e.g., class labels.
> > > > * The current study allows outputs outside the domain, e.g., soft labels.
> > > >
> > > > *4. The Usage of Relabeling*
> > > > * The suggested relabelings by [1] are assessed by the experts and only wrong labels are corrected.
> > > > * The current study do not require human assessments and the new label are directly used for the new training.
> > > >
> > > > Here, I would like to point out that the differences above are not significant when evaluating the technical novelty of the paper.
> > > > Indeed, *1. The Objective of Relabeling*, *2. Target of Relabeling*, and *4. The Usage of Relabeling* have nothing to do with the algorithms in the current paper and  [1]. These are all up to the users.
> > > >
> > > > For example, in *1. The Objective of Relabeling* and *2. Target of Relabeling*, one can treat the validation set as the untrusted item in [1], and set trusted item to null. This will result in the same objective function, i.e., simply improving the validation loss, while allowing all labels to be changed.
> > > > That is, the problem setup of [1] includes the problem of the current paper as its special case.
> > > >
> > > > *4. The Usage of Relabeling* is also completely independent matters from the algorithms.
> > > > The users have a choice of not assessing the suggested relabelings, and directly using them for the new trainig.
> > > >
> > > > ## Relabeling Function and Optimization
> > > > * [1] uses bi-level optimization.
> > > > * The current paper uses the influence function.
> > > >
> > > > [Ref1] used implicit differentiation for bi-level optimization.
> > > > Here, I would like to remind that implicit differentiation includes influence function as its special case.
> > > > See e.g.,  [Ref2], for the detail.
> > > > * [Ref2] Optimizing Millions of Hyperparameters by Implicit Differentiation, AISTATS 202
> > > >
> > > > Moreover, I think the statement below seems to be too strong.
> > > > > we propose an explicit relabeling function which has nothing to do with gradient, validation loss or even influence function
> > > >
> > > > The selection of relabeled instances depend on validation loss and influence function.
> > > > In this regard, the entire relabeling procedure depends on the gradient (i.e., influence function on the validation set as a special case of implicit differentiation).
> > > >
> > > > I do agree that the way the amount of relabeling is determined is different and would be a novelty of the paper.
> > > >
> > > > ## Theoretical Guarantee
> > > > * [1] does not provide any theory.
> > > > * The current paper provide the theory on reduction of validation loss.
> > > >
> > > > I confirm this is true.
> > > > I also agree that proving the reduction of the validation loss for the specific relabeling function under consideration would be a novelty of the paper.
> > > > However, I think it would not appropriate to underestimate the contribution of [Ref1] just because they did not make the trivial claim.
> > > > We will not say explicitly in the paper that "the loss reduces because we use gradient descent".
> > > > This is a trivial fact in the field (as long as the step size is sufficiently small).

---

> > > > > ### Author Response · Authors · 2021-11-19
> > > > > **Thank you very much, some clarifications**
> > > > >
> > > > > Thank you very much for your further detailed feedback! It is a delight to know that we have agreed on some of the novelty of our method, e.g., Lemma, Theorem and the relabeling function. We address your remaining concerns below.
> > > > >
> > > > >
> > > > >
> > > > > Please feel free to ask any further questions if our responses are still insufficient.
> > > > >
> > > > >
> > > > >
> > > > > # Target
> > > > >
> > > > > What we want to clarify is that we are not saying [1] could not relabel wrong labels for improving model performance, but that it is not easy to decide the amount of relabeled instances that can lead to better model performance. We agree with the reviewer that when adapting [1] to our problem setting, we could simply allow all labels to be changed and then measure the change in the test loss. However, the model performance in [1] will be severely affected by the predefined budget b which manually controls the amount of relabeled (detected) data. For example, if we only have 10 wrong labels in the training set but set b to be 100, then we may end up with 90 noisy samples after being relabeled in the worst situation. Following your suggestion, we have run the code published in [1] and found that there is a threshold to control the maximum number of iterations, but it is still possible that more than 10 samples are relabeled before the max iteration. The situation will be even worse when the number of wrong labels increases. For example, if we have 300 wrong labels in the training set but set b to be 100, we will leave the remaining 200 wrong labels unchanged. We believe that [1] is an excellent bug detection framework with a novel optimization scheme. However, applying [1] on relabeling all the training samples is not suitable since budget b should be finely set. We have added the comparison experiments in the experiment part to verify our claims.
> > > > >
> > > > > # Novelty
> > > > > > However, the way the influential instances are selected is essentially the same as [Ref1] because the influence function is one specific example of the implicit differentiation used in the bi-level optimization.
> > > > >
> > > > > We agree with the reviewer that the influence function is one specific example of the implicit differentiation used in the bi-level optimization, However, we would like to clarify that our contribution is not on how to select the influential instances, but on how to relabel the identified harmful instances which can improve the model performance. As mentioned in the previous rebuttal round, how to propose an explicit relabeling function which has a theoretical guarantee on the model performance is the main technical challenge. Besides, to the best of our knowledge, we are the first one to propose the influence-based relabeling approach which could theoretically achieve the lower test loss than the existing influence-based resampling approach.
> > > > >
> > > > >
> > > > >
> > > > > > The labels of instances that are influential for the validation loss are updated to reduce the validation loss, i.e., the labels are *trained* (although the amount of relabeling is determined by the training pair and the current model).
> > > > >
> > > > >
> > > > >
> > > > > We agree with the reviewer that the selection of harmful instances depends on validation loss and influence function in RDIA. However, we would like to illustrate that our relabeling function is not a trainable parameter, and hence it is independent with the validation loss or influence function (As we claimed in the first period of rebuttal, this is the biggest difference between our relabeling function and [1]). An important fact is that our relabeling function can be directly incorporated with other harmful data selection methods. **We have already proposed the RDIA-LS (Appendix F) that uses training loss to identify harmful samples (rather than influence function) and applies our relabeling function to relabel them, achieving good performance on deep models. In this situation, we do not even have a validation set and our relabeling function still works.**
> > > > > The reason is that our relabeling function is not trainable, but only uses the current model’s predictions to relabel harmful samples. This contribution has also been appraised by reviewer 136B and UYUF.

---

> > > > > > ### Author Response · Authors · 2021-11-19
> > > > > > **More experiments**
> > > > > >
> > > > > > We would much appreciate it if you could kindly comment on our newly added experiments below. For now, we do not have enough time to finish all the experiments. We will try our best to compare [1] and our approach on more datasets for a comprehensive comparison.
> > > > > >
> > > > > >
> > > > > >
> > > > > > Here we use the code published in the original paper (Matlab) and do the experiment on two relatively small datasets  “Breast-cancer” and “Diabetes”. Specifically, we follow our experimental settings and treat the validation set as trusted items with confidence c=100. We first use [1] to relabel the wrong labels (We use all the original hyperparameters in [1] such as the max iterations.)
> > > > > >
> > > > > > We finely set the budget b which controls the amount of relabeled samples and then retrain the model (logistic regression) with the relabeled training set and report the Logloss in an additional out-of-sample test set which ensures we do not utilize any information of the test data in advance. We use n to represent the amount of training samples.
> > > > > >
> > > > > >
> > > > > >
> > > > > > | Dataset:  Cancer |  ERM   | [1]  with b=0.1n | [1]  with b=0.5n | [1]  with b= n |  RDIA  |
> > > > > > | :--------------: | :----: | :--------------: | :--------------: | :------------: | :----: |
> > > > > > |      Clean       | 0.0914 |      0.0885      |      0.0955      |     0.1557     | 0.0649 |
> > > > > > | 20%  noisy rate  | 0.1682 |      0.0972      |      0.1393      |     0.1869     | 0.0754 |
> > > > > > | 40%  noisy rate  | 0.5401 |      0.2574      |      0.1782      |     0.1849     | 0.0871 |
> > > > > > | 60%  noisy rate  | 0.8839 |      0.3604      |      0.3887      |     0.2705     | 0.0898 |
> > > > > >
> > > > > >
> > > > > > | Dataset: Diabetes |  ERM   | [1]  with b=0.1n | [1]  with b=0.5n | [1]  with b= n |  RDIA  |
> > > > > > | :---------------: | :----: | :--------------: | :--------------: | :------------: | :----: |
> > > > > > |       Clean       | 0.5461 |      0.5267      |      0.5701      |     0.5976     | 0.4920 |
> > > > > > |  20%  noisy rate  | 0.5649 |      0.5402      |      0.5495      |     0.6223     | 0.5131 |
> > > > > > |  40%  noisy rate  | 0.6355 |      0.5962      |      0.5711      |     0.6325     | 0.5663 |
> > > > > > |  60%  noisy rate  | 0.7685 |      0.6721      |      0.5985      |     0.6560     | 0.5925 |
> > > > > >
> > > > > >
> > > > > >
> > > > > >
> > > > > >
> > > > > >
> > > > > > We have found that b highly affects the model performance. Nevertheless, our approach consistently outperforms [1] since we use influence function to only relabel the harmful samples and update their labels at one shot. Besides, [1] aims to find the smallest change in label and uses $\gamma$ to control the magnitude of label change (Equation (9) in [1]). This would also restrict the model in handling data with high noise ratios.
> > > > > >
> > > > > >
> > > > > >
> > > > > > We hope we have addressed your questions; please let us know if there are any additional concerns.

---

> > > > > > > ### Author Response · Authors · 2021-11-23
> > > > > > > **Experiments on larger dataset**
> > > > > > >
> > > > > > >
> > > > > > >
> > > > > > > | Dataset:   MNIST |  ERM   |   [1]   with b=0.1n    |   [1]   with b=0.5n    | [1]   with b= n |    RDIA    |
> > > > > > > | :--------------: | :----: | :--------------------: | :--------------------: | :-------------: | :--------: |
> > > > > > > |      Clean       | 0.0245 | $\underline{0.0228  }$ |         0.0239         |     0.0347      | **0.0207** |
> > > > > > > |  20% noisy rate  | 0.2567 | $\underline{0.1336  }$ |         0.2317         |     0.2543      | **0.0386** |
> > > > > > > |  40% noisy rate  | 0.5296 |         0.4135         | $\underline{0.3877  }$ |     0.5304      | **0.0413** |
> > > > > > > |  60% noisy rate  | 0.9164 |         0.8860         | $\underline{0.5985  }$ |     0.6752      | **0.0503** |
> > > > > > >
> > > > > > > In addition to the two small datasets provided in our earlier response, the results on the larger dataset MNIST with Logistic regression are collected in the table below.
> > > > > > >
> > > > > > >
> > > > > > >
> > > > > > > From the results over the three datasets, we obtain the same conclusions as follows. First, the performance of [1] is greatly affected by the budget b which controls the amount of relabeled samples. In practice, it may not be desirable to ask experts to decide an appropriate value of b for every dataset. Note that the bi-level optimization is computation-intensive and hence it is time consuming to treat b as a hyperparameter to be searched. For instance, the influence function only takes 4.9 seconds to identify all the harmful samples in the MNIST while [1] takes more than 2 hours to detect and relabel noisy samples with a predefined budget b.
> > > > > > >
> > > > > > >
> > > > > > >
> > > > > > > Second, our RDIA approach consistently outperforms [1] on all the three datasets and different noise ratios. This is because the amount of relabeled samples selected by the influence function is close to the real number of wrong labels. This is also the reason why the influence-based resampling approach Dropout [2] which simply drop all the identified harmful samples still consistently outperforms [1].
> > > > > > >
> > > > > > > Third, when the noise ratio increases, the advantage of RDIA is more significant than [1]. The main reason is that [1] aims to find the smallest change in label and restrict the magnitude of label change (Equation (9) in [1]) while our approach relabels all the harmful samples identified by the influence function towards better model performance.
> > > > > > >
> > > > > > > To sum up, due to different objectives, [1] mainly focuses on identifying the preferred total number b of flagged bugs and hence is inappropriate to resolve all the biases in a training set which is the focus of our paper. In this regard, the chosen comparison methods are the state-of-the-art influence-based reweighting approaches [2,3,4] which upweight/downweight training instances via influence function.
> > > > > > >
> > > > > > >
> > > > > > >
> > > > > > > We hope we have clarified the difference between our approach and [1]. We sincerely thank the reviewer again for reviewing our paper and giving further feedbacks in the rebuttal round. We are very willing to explain more if there are still any concerns.
> > > > > > >
> > > > > > >
> > > > > > >
> > > > > > > [1] Training Set Debugging Using Trusted Items, AAAI 2018.
> > > > > > >
> > > > > > > [2] Data dropout: Optimizing training data for convolutional neural networks, ICTAI 2018.
> > > > > > >
> > > > > > > [3] Less Is Better: Unweighted Data Subsampling via Influence Function, AAAI2020.
> > > > > > >
> > > > > > > [4] Optimal Subsampling with Influence Functions. NIPS 2018.

---

> > > > > > > ### Comment · Reviewer_FfyB · 2021-11-24
> > > > > > > **Re: More experiments**
> > > > > > >
> > > > > > > I would like to thank the authors for the additional experiments.
> > > > > > >
> > > > > > > I also conducted the experiment on breast-cancer by myself (see my next post).
> > > > > > > I confirmed that RDIA works well.
> > > > > > > However, I observed that there can be a discrepancy between the loss and accuracy.
> > > > > > > When I looked at the results in terms of test accuracy, One-Step GD update could perform well in the low noise regime.
> > > > > > > Based on this observation, I wonder whether the reported improvements in loss imply improvements in accuracy as well.
> > > > > > >
> > > > > > > *Test Accuracy*
> > > > > > >
> > > > > > > |  Noise Ratio   |   ERM |   One-Step GD |   RDIA |
> > > > > > > |----:|------:|--------------:|-------:|
> > > > > > > | 0   | 0.95  |         0.96  |  0.923 |
> > > > > > > | 0.1 | 0.929 |         0.968 |  0.945 |
> > > > > > > | 0.2 | 0.872 |         0.951 |  0.947 |
> > > > > > > | 0.3 | 0.728 |         0.944 |  0.946 |
> > > > > > > | 0.4 | 0.497 |         0.897 |  0.949 |
> > > > > > > | 0.5 | 0.354 |         0.762 |  0.952 |
> > > > > > > | 0.6 | 0.241 |         0.782 |  0.948 |
> > > > > > > | 0.7 | 0.141 |         0.875 |  0.953 |
> > > > > > > | 0.8 | 0.078 |         0.931 |  0.955 |
> > > > > > > | 0.9 | 0.053 |         0.95  |  0.952 |
> > > > > > >
> > > > > > > *Test Loss*
> > > > > > >
> > > > > > > |  Noise Ratio   |      ERM |   One-Step GD |     RDIA |
> > > > > > > |----:|---------:|--------------:|---------:|
> > > > > > > | 0   | 0.141025 |      0.161033 | 0.239982 |
> > > > > > > | 0.1 | 0.237151 |      0.24697  | 0.188336 |
> > > > > > > | 0.2 | 0.345921 |      0.347718 | 0.172467 |
> > > > > > > | 0.3 | 0.453018 |      0.451272 | 0.180556 |
> > > > > > > | 0.4 | 0.56437  |      0.557865 | 0.186278 |
> > > > > > > | 0.5 | 0.698697 |      0.63033  | 0.186627 |
> > > > > > > | 0.6 | 0.876071 |      0.590184 | 0.181963 |
> > > > > > > | 0.7 | 1.08732  |      0.492453 | 0.166516 |
> > > > > > > | 0.8 | 1.44151  |      0.362804 | 0.163885 |
> > > > > > > | 0.9 | 2.0108   |      0.265325 | 0.153742 |

---

> > > > > > > > ### Comment · Reviewer_FfyB · 2021-11-24
> > > > > > > > **Re: More experiments**
> > > > > > > >
> > > > > > > > My code is as follows.
> > > > > > > >
> > > > > > > > ```
> > > > > > > > import numpy as np
> > > > > > > > import pandas as pd
> > > > > > > > from sklearn import datasets
> > > > > > > > from sklearn.model_selection import train_test_split
> > > > > > > > from sklearn.preprocessing import StandardScaler
> > > > > > > > from IPython.display import display
> > > > > > > >
> > > > > > > > def sigmoid(u):
> > > > > > > >     return 1.0 / (1.0 + np.exp(-u))
> > > > > > > >
> > > > > > > > def logloss(w, b, x, y):
> > > > > > > >     p = sigmoid(x.dot(w) + b)
> > > > > > > >     return - (y.dot(np.log(p)) + (1 - y).dot(np.log(1 - p))) / y.size
> > > > > > > >
> > > > > > > > def logreg(x, y, lam=0.1, lr=0.01, max_itr=100):
> > > > > > > >     num, dim = x.shape
> > > > > > > >     xo = np.concatenate([x, np.ones((num, 1))], axis=1)
> > > > > > > >     w = np.zeros(dim+1)
> > > > > > > >     for itr in range(max_itr):
> > > > > > > >         p = sigmoid(xo.dot(w))
> > > > > > > >         g = (p - y).dot(xo) / num
> > > > > > > >         g[:-1] = g[:-1] + lam * w[:-1]
> > > > > > > >         w = w - lr * g
> > > > > > > >     return w[:-1], w[-1]
> > > > > > > >
> > > > > > > > # experiment
> > > > > > > > lam = 1e-2
> > > > > > > > acc, loss = [], []
> > > > > > > > nr = np.linspace(0, 0.9, 10)
> > > > > > > > for noise_rate in nr:
> > > > > > > >     acc_n, loss_n = [], []
> > > > > > > >     for seed in range(10):
> > > > > > > >
> > > > > > > >         # breast cancer data
> > > > > > > >         # train / val / test = 300 / 169 / 100
> > > > > > > >         x, y = datasets.load_breast_cancer(return_X_y=True, as_frame=False)
> > > > > > > >         x, xte, y, yte = train_test_split(x, y, test_size=269, random_state=seed)
> > > > > > > >         xval, xte, yval, yte = train_test_split(xte, yte, test_size=100, random_state=seed)
> > > > > > > >
> > > > > > > >         # normalize features
> > > > > > > >         scaler = StandardScaler().fit(x)
> > > > > > > >         x = scaler.transform(x)
> > > > > > > >         xval = scaler.transform(xval)
> > > > > > > >         xte = scaler.transform(xte)
> > > > > > > >
> > > > > > > >         # noisy label in training
> > > > > > > >         np.random.seed(seed)
> > > > > > > >         flip = np.random.rand(y.size) < noise_rate
> > > > > > > >         y = np.logical_xor(y, flip).astype(int)
> > > > > > > >
> > > > > > > >         # fit logreg
> > > > > > > >         w, b = logreg(x, y, max_itr=500, lam=lam)
> > > > > > > >
> > > > > > > >         # test accuracy / loss
> > > > > > > >         zte = (xte.dot(w) + b > 0.5).astype(int)
> > > > > > > >         acc_te1 = np.mean(yte == zte)
> > > > > > > >         loss_te1 = logloss(w, b, xte, yte)
> > > > > > > >
> > > > > > > >         # influence function
> > > > > > > >         num, dim = x.shape
> > > > > > > >         xo = np.concatenate([x, np.ones((num, 1))], axis=1)
> > > > > > > >         p = sigmoid(x.dot(w) + b)
> > > > > > > >         H = xo.T.dot((p * (1 - p))[:, np.newaxis] * xo) / num
> > > > > > > >         H = 0.5 * (H + H.T) + np.diag([lam] * dim + [0]) # Hessian
> > > > > > > >         g = ((p - y)[:, np.newaxis] * xo) / num # gradient
> > > > > > > >         f = - np.linalg.solve(H, g.T).T # influence function
> > > > > > > >
> > > > > > > >         # label gradient over validation loss
> > > > > > > >         xv = np.concatenate([xval, np.ones((xval.shape[0], 1))], axis=1)
> > > > > > > >         q = sigmoid(xval.dot(w) + b)
> > > > > > > >         gv = (q - yval).dot(xv) / xval.shape[0] # gradient of validation loss
> > > > > > > >         gy = f.dot(gv) * (np.log(p) - np.log(1 - p)) # label gradient
> > > > > > > >
> > > > > > > >         # label correction by one-step gradient descent
> > > > > > > >         ynew = y - 1e+3 * gy # step size of GD = 1e+3
> > > > > > > >         ynew = np.minimum(1, np.maximum(0, ynew)) # clip label
> > > > > > > >         wnew, bnew = logreg(x, ynew, max_itr=500, lam=lam)
> > > > > > > >
> > > > > > > >         # test accuracy / loss after label correction
> > > > > > > >         zte = (xte.dot(wnew) + bnew > 0).astype(int)
> > > > > > > >         acc_te2 = np.mean(yte == zte)
> > > > > > > >         loss_te2 = logloss(wnew, bnew, xte, yte)
> > > > > > > >
> > > > > > > >         # label correction by RDIA
> > > > > > > >         ynew = y.copy()
> > > > > > > >         ynew[f.dot(gv) > 0] = 1 - ynew[f.dot(gv) > 0] # flip label
> > > > > > > >         wnew, bnew = logreg(x, ynew, max_itr=500, lam=lam)
> > > > > > > >
> > > > > > > >         # test accuracy / loss after label correction
> > > > > > > >         zte = (xte.dot(wnew) + bnew > 0).astype(int)
> > > > > > > >         acc_te3 = np.mean(yte == zte)
> > > > > > > >         loss_te3 = logloss(wnew, bnew, xte, yte)
> > > > > > > >
> > > > > > > >         acc_n.append((acc_te1, acc_te2, acc_te3))
> > > > > > > >         loss_n.append((loss_te1, loss_te2, loss_te3))
> > > > > > > >
> > > > > > > >     acc.append(acc_n)
> > > > > > > >     loss.append(loss_n)
> > > > > > > > acc = np.array(acc)
> > > > > > > > loss = np.array(loss)
> > > > > > > >
> > > > > > > > acc = pd.DataFrame(np.mean(acc, axis=1))
> > > > > > > > acc.columns = ['ERM', 'One-Step GD', 'RDIA']
> > > > > > > > acc.index = nr
> > > > > > > > loss = pd.DataFrame(np.mean(loss, axis=1))
> > > > > > > > loss.columns = ['ERM', 'One-Step GD', 'RDIA']
> > > > > > > > loss.index = nr
> > > > > > > > print('Test Accuracy')
> > > > > > > > display(acc)
> > > > > > > > print('Test Loss')
> > > > > > > > display(loss)
> > > > > > > > ```

---

> > > > > > > > ### Author Response · Authors · 2021-11-25
> > > > > > > > **Reply to Reviewer FfyB**
> > > > > > > >
> > > > > > > > Thank you very much for your further detailed feedback. We would like to address your concerns below.
> > > > > > > >
> > > > > > > >
> > > > > > > >
> > > > > > > > First of all, we thank the reviewer again for the patient response and additional experiments. According to the provided code, we found there is a  mistake in **Line 83 and Line 93**:
> > > > > > > >
> > > > > > > > ```
> > > > > > > > Line 83&93: zte = (xte.dot(wnew) + bnew > 0).astype(int)
> > > > > > > > ```
> > > > > > > >
> > > > > > > > Generally speaking, we treat the model prediction as the probability of the label belonging to 1. Hence, we typically do not set the label to 1 if the model prediction is only bigger than 0. We believe Line 83 and Line 93 in the provided code need to be consistent with  **Line 58**, i.e.,
> > > > > > > >
> > > > > > > > ```
> > > > > > > > Line 58 : zte = (xte.dot(wnew) + bnew > 0.5).astype(int)
> > > > > > > > ```
> > > > > > > >
> > > > > > > > For Line **89**, we use a hyperparameter $\alpha$ to control the amount of the relabeled samples to tolerate the estimation error of influence function which may result in the wrong identification of harmful training samples (see Section 3.2). This is similar to the existing influence-based reweighting approach [3] which uses a hyperparameter to control the sampling ratio. Note that the setting of $\alpha$ and its effect on model performance are provided in Section 3.2 and Section 5.3, respectively. For simplicity, here we set $\alpha$ to 0.001 and have the updated Line 89 as follows:
> > > > > > > >
> > > > > > > > ```
> > > > > > > > Line 89 (updated): ynew[f.dot(gv) > 0.001] = 1 - ynew[f.dot(gv) > 0.001] # flip label
> > > > > > > > ```
> > > > > > > >
> > > > > > > > After changing Line 83,89 and 93, we run the code and get the result below:
> > > > > > > >
> > > > > > > > Test accuracy
> > > > > > > >
> > > > > > > > | Noise Ratio | ERM   | One-Step GD | RDIA      |
> > > > > > > > | ----------- | ----- | ----------- | --------- |
> > > > > > > > | 0.0         | 0.950 | 0.947       | **0.950** |
> > > > > > > > | 0.1         | 0.929 | 0.935       | 0.947     |
> > > > > > > > | 0.2         | 0.872 | 0.873       | 0.944     |
> > > > > > > > | 0.3         | 0.728 | 0.745       | 0.939     |
> > > > > > > > | 0.4         | 0.497 | 0.458       | 0.943     |
> > > > > > > > | 0.5         | 0.354 | 0.375       | 0.941     |
> > > > > > > > | 0.6         | 0.241 | 0.448       | 0.939     |
> > > > > > > > | 0.7         | 0.141 | 0.652       | 0.935     |
> > > > > > > > | 0.8         | 0.078 | 0.848       | 0.931     |
> > > > > > > > | 0.9         | 0.053 | 0.908       | 0.922     |
> > > > > > > >
> > > > > > > > Test loss
> > > > > > > >
> > > > > > > > | Noise Ratio | ERM      | One-Step GD | RDIA    |
> > > > > > > > | ----------- | -------- | ----------- | ------- |
> > > > > > > > | 0.0         | 0.141025 | 0.161033    | 0.13819 |
> > > > > > > > | 0.1         | 0.237151 | 0.246970    | 0.14611 |
> > > > > > > > | 0.2         | 0.345921 | 0.347718    | 0.15734 |
> > > > > > > > | 0.3         | 0.453018 | 0.451272    | 0.17178 |
> > > > > > > > | 0.4         | 0.564370 | 0.557865    | 0.18598 |
> > > > > > > > | 0.5         | 0.698697 | 0.630330    | 0.19716 |
> > > > > > > > | 0.6         | 0.876071 | 0.590184    | 0.20312 |
> > > > > > > > | 0.7         | 1.087316 | 0.492453    | 0.19651 |
> > > > > > > > | 0.8         | 1.441506 | 0.362804    | 0.19738 |
> > > > > > > > | 0.9         | 2.010796 | 0.265325    | 0.24127 |
> > > > > > > >
> > > > > > > > We can see that RDIA consistently performs better than One-Step GD both in test loss and test accuracy.
> > > > > > > >
> > > > > > > >
> > > > > > > >
> > > > > > > > As for the metrics used in the paper, we follow the influence-based resampling approach [3] and measure the test loss since our objective is to optimize the test loss via influence analysis. Therefore, we compare the test loss between RDIA and other influence-based approach.
> > > > > > > >
> > > > > > > >
> > > > > > > >
> > > > > > > > In what follows, we further provide the accuracy results on four datasets with logistic regression to show that our approach can still effectively improve the accuracy. Since there are some class imbalanced datasets (i.e., diabetes), we report the test loss/test **AUC** below
> > > > > > > >
> > > > > > > >
> > > > > > > >
> > > > > > > > Clean
> > > > > > > >
> > > > > > > > | **Dataset** | **MNIST**     | **Real-sim**   | **Diabetes**   | **Cancer**  |
> > > > > > > > | ----------- | ------------- | -------------- | -------------- | ----------- |
> > > > > > > > | ERM         | 0.0248/0.9921 | 0.2606/0.9883  | 0.5169/0.8679  | 0.0914/1.00 |
> > > > > > > > | UIDS         | 0.0238/0.9922 | 0.2607/0.9884  | 0.5068/ 0.8682 | 0.0786/1.00 |
> > > > > > > > | RDIA        | 0.0209/0.9926 | 0.2585/ 0.9875 | 0.4920/ 0.8687 | 0.0649/1.00 |
> > > > > > > >
> > > > > > > > Noise ratio = 0.4
> > > > > > > >
> > > > > > > > | **Dataset** | **MNIST**     | **Real-sim**   | **Diabetes**   | **Cancer**    |
> > > > > > > > | ----------- | ------------- | -------------- | -------------- | ------------- |
> > > > > > > > | ERM         | 0.5277/0.9646 | 0.5869/0.9640  | 0.6730/0.7801  | 0.5646/0.9679 |
> > > > > > > > | UIDS        | 0.4699/0.9836 | 0.5414/0.9785  | 0.6160/ 0.7974 | 0.3739/1.00   |
> > > > > > > > | RDIA        | 0.0434/0.9991 | 0.2714/ 0.9820 | 0.5350/ 0.8170 | 0.0871/1.00   |
> > > > > > > >
> > > > > > > > Noise ratio = 0.8
> > > > > > > >
> > > > > > > > | Dataset | MNIST         | Real-sim       | Diabetes       | Cancer        |
> > > > > > > > | ------- | ------------- | -------------- | -------------- | ------------- |
> > > > > > > > | ERM     | 1.611/0.0023  | 1.1772/0.0112  | 0.8254/0.1448  | 1.3866/0.0003 |
> > > > > > > > | UIDS     | 1.148/ 0.0105 | 0.9252/0.0210  | 0.7310/ 0.5953 | 1.1203/0.009  |
> > > > > > > > | RDIA    | 0.0406/0.9990 | 0.2672/ 0.9785 | 0.5860/ 0.8078 | 0.0826/1.00   |
> > > > > > > >
> > > > > > > > Based on the observations, we have found that the test loss results are consistent with test AUC results. When the noise ratio increases, RDIA outperforms the baselines by a large margin in both test loss and test AUC.
> > > > > > > >
> > > > > > > >
> > > > > > > >
> > > > > > > > We hope we have addressed your questions; please let us know if there are any additional concerns.

---

> > > > > > > > > ### Comment · Reviewer_FfyB · 2021-11-25
> > > > > > > > > **Re: Reply to Reviewer FfyB**
> > > > > > > > >
> > > > > > > > > Thank you for pointing out the bug.
> > > > > > > > > Actually, the threshold 0.5 in Line 58 was the bug, while the threshold 0 in Line 83 and Line 93 were correct.
> > > > > > > > > I also implemented the suggest thresholding with alpha = 0.001.
> > > > > > > > > Then, I observed that the scores of RDIA improved (see the end of this post).
> > > > > > > > >
> > > > > > > > > At this point, I confirmed that RDIA does better than the One-Step GD update of [Ref1].
> > > > > > > > > I will update my score according to this result, and with a strong expectation to the authors for **referring [Ref1] appropriately in the main body of the paper**.
> > > > > > > > > As I mentioned in my review, this is not the first study considering relabeling based on the influence function-like technique.
> > > > > > > > > To my knowledge, [Ref1] is the first work in this direction (even if [Ref1] does not describe the fact explicitly).
> > > > > > > > > As I demonstrated in my code, the simplest version of [Ref1] with only one-step update (and without any human intervention) does good job.
> > > > > > > > > **I strongly expect the authors to pay certain respect to [Ref1] and do not underestimate their technical contribution**.
> > > > > > > > > For example, the current writing in Appendix H seems to be not appropriate.
> > > > > > > > > Even without human intervention, DUTI [Ref1] can do the same (i.e., relabel the training bias towards better model performance) as the proposed approach (but with slightly inferior performance).
> > > > > > > > > > (2) The learning task of DUTI is to debug the training instances which may contain the wrong labels and predict the true labels. While the target of our approach is to relabel the training bias towards better model performance. In this way, only our approach could relabel the biased training samples with correct labels towards better model performance.
> > > > > > > > >
> > > > > > > > > *Test Accuracy*
> > > > > > > > >
> > > > > > > > > |     |   ERM |   One-Step GD |   RDIA |
> > > > > > > > > |----:|------:|--------------:|-------:|
> > > > > > > > > | 0   | 0.959 |         0.96  |  0.961 |
> > > > > > > > > | 0.1 | 0.96  |         0.968 |  0.962 |
> > > > > > > > > | 0.2 | 0.953 |         0.951 |  0.959 |
> > > > > > > > > | 0.3 | 0.931 |         0.944 |  0.957 |
> > > > > > > > > | 0.4 | 0.842 |         0.897 |  0.958 |
> > > > > > > > > | 0.5 | 0.563 |         0.762 |  0.958 |
> > > > > > > > > | 0.6 | 0.178 |         0.782 |  0.953 |
> > > > > > > > > | 0.7 | 0.075 |         0.875 |  0.953 |
> > > > > > > > > | 0.8 | 0.05  |         0.931 |  0.949 |
> > > > > > > > > | 0.9 | 0.037 |         0.95  |  0.94  |
> > > > > > > > >
> > > > > > > > > *Test Loss*
> > > > > > > > >
> > > > > > > > > |     |      ERM |   One-Step GD |     RDIA |
> > > > > > > > > |----:|---------:|--------------:|---------:|
> > > > > > > > > | 0   | 0.141025 |      0.161033 | 0.13819  |
> > > > > > > > > | 0.1 | 0.237151 |      0.24697  | 0.146115 |
> > > > > > > > > | 0.2 | 0.345921 |      0.347718 | 0.157343 |
> > > > > > > > > | 0.3 | 0.453018 |      0.451272 | 0.171787 |
> > > > > > > > > | 0.4 | 0.56437  |      0.557865 | 0.185989 |
> > > > > > > > > | 0.5 | 0.698697 |      0.63033  | 0.197167 |
> > > > > > > > > | 0.6 | 0.876071 |      0.590184 | 0.203124 |
> > > > > > > > > | 0.7 | 1.08732  |      0.492453 | 0.196512 |
> > > > > > > > > | 0.8 | 1.44151  |      0.362804 | 0.197389 |
> > > > > > > > > | 0.9 | 2.0108   |      0.265325 | 0.241273 |

---

> > > > > > > > > > ### Author Response · Authors · 2021-11-26
> > > > > > > > > > **Thanks for your further comments**
> > > > > > > > > >
> > > > > > > > > > Dear reviewer,
> > > > > > > > > >
> > > > > > > > > > We appreciate the response from the reviewer. It is good to see we are now in sync about the effectiveness of RDIA.
> > > > > > > > > >
> > > > > > > > > > Following your constructive suggestion, we would like to describe [Ref1] appropriately in the main body of the paper and include the additional comparison results in the Appendix. Further, we will improve Appendix H to discuss the relation between [Ref1] and our proposed method in a better way. Basically, the DUTI algorithm in [Ref1] learns a relabeling function by bi-level optimization, which has shown to be effective in identifying mislabeled training items and improving learning according to our experiments.
> > > > > > > > > >
> > > > > > > > > > Once again, thank you very much for your further comments.
> > > > > > > > > >
> > > > > > > > > > Best，
> > > > > > > > > >
> > > > > > > > > > The authors

---

> > > > ### Comment · Reviewer_FfyB · 2021-11-17
> > > > **Reply (2/3)**
> > > >
> > > > ## Experiments
> > > > > Since the experimental settings and metrics are totally different in two works, we could not conduct the comparison experiments.
> > > >
> > > > Please remind that I am not asking to compare the results based on the metrics (detection of wrong labels) used in [Ref1].
> > > > What I expect is to compare [Ref1] in the metrics (loss improvement) used in the current paper because the method of [Ref1] can be used for the same purpose as the current paper under the same problem setup.
> > > > How much improvements of the loss would be observed by following the suggested relabeling by [Ref1]?
> > > > (under the setting where all the validation set are untrusted items, there are no trusted items, and no assessment by the experts)
> > > >
> > > > ## "our equation (10) is not identical to the above formula"
> > > >
> > > > Let me apologize that the equations are slightly incorrect.
> > > > The correct one is
> > > >
> > > > $$
> > > > \frac{dL(Q; \hat{\theta}\_\delta)}{d\delta\_i} = \nabla\_\theta L(Q; \hat{\theta}\_\delta) \frac{d\hat{\theta}\_\delta}{d\delta\_i} = -\frac{1}{N}\nabla\_\theta L(Q; \hat{\theta}\_\delta)^\top H\_{\hat{\theta}\_\delta}^{-1} \frac{\partial \nabla_\theta l\_i(z_{i\delta_i}, \hat{\theta}\_\delta)}{\partial \delta\_i} .
> > > > $$
> > > >
> > > > which will lead to
> > > >
> > > > $$
> > > > L(Q; \hat{\theta}\_{\delta}) - L(Q; \hat{\theta}\_0) \approx \epsilon \left. \frac{dL(Q; \hat{\theta}\_\delta)}{d\delta\_i} \right|\_{\delta\_i = 0}
> > > > $$
> > > > $$
> > > > = \frac{\epsilon}{N} \nabla\_\theta L(Q; \hat{\theta}\_0)^\top H\_{\hat{\theta}\_0}^{-1} \frac{\partial \nabla_\theta l\_i(z\_{i\delta\_i}, \theta_0)}{\partial \delta\_i}
> > > > $$
> > > > $$
> > > > \approx \frac{1}{N}\nabla\_\theta L(Q; \hat{\theta}\_0)^\top H\_{\hat{\theta}\_0}^{-1} \nabla_\theta(l\_i(z\_{i \epsilon}, \theta) - l\_i(\theta)) .
> > > > $$

---

> > > > ### Comment · Reviewer_FfyB · 2021-11-17
> > > > **Reply (3/3)**
> > > >
> > > > ## Lemma1 and Theorem1 are non-trivial.
> > > >
> > > > I agree that the proposed specific relabeling function and its validity are nontrivial.
> > > > However, the following statement is questionable to me.
> > > > > our proposed relabeling function $\delta$ is not a trainable parameter
> > > > The labels of instances that are influential for the validation loss are updated to reduce the validation loss, i.e., the labels are *trained* (although the amount of relabeling is determined by the training pair and the current model).
> > > >
> > > > Overall, I confirm there are a few differences from [Ref1] in the current paper.
> > > > In particular, degerming the amount of relabeling only by the training pair and the current model seems to be novel and interesting.
> > > > However, the way the influential instances are selected is essentially the same as [Ref1] because the influence function is one specific example of the implicit differentiation used in the bi-level optimization.

---

> ### Author Response · Authors · 2021-11-22
> **Thanks for your feedbacks**
>
> Dear reviewer,
>
> Thank you very much for your further feedbacks. For your remaining concerns about the novelty of our approach, we have provided more clarifications as well as the supporting experiments. Please feel free to ask any further questions.
>
> Thanks.

---

### Official Review · Reviewer_136B · 2021-10-31

**Correctness:** 4
**Technical Novelty And Significance:** 4
**Empirical Novelty And Significance:** 4
**Recommendation:** 6
**Confidence:** 5

**Main Review:**

The paper uses  influence functions from robust statistics to first identify harmful instances and then relabel them based on a novel relabeling function using influence approximation. Influence estimations have been widely used to identify harmful instances or understand the impact of training samples. This paper goes one step further and uses this analysis to relabel the instances in order to achieve better generalization and lower test error. While the technical novelty is limited as the proposed formulations are extensions of existing influence functions and application is interesting. The authors use their relabeling strategy on multiple datasets and models (including deep models).

Pros:

(1) The paper does a focused job of using influence function for identifying harmful examples and fixing them by relabeling. While in the recent times there are papers on influence functions to identify harmful instances, this paper does a very good comprehensive and focused study compared to others.

(2) The paper is well-written and easy to follow. Related works is well laid out and well covered.

(3) Experimental section is complete with experiments on deep models and the proposed RDIA-LS.  The authors acknowledge the limitations of RDIA on deep models due to erroneous influence estimates for deep models and provide a workaround for it in the Appendix. This is an improvement from the earlier versions of the paper (from a previous conference). I would definitely like to highlight this section in the main paper as it’s important for all practical purposes.


Cons:

(1) The technique of using influence function for identifying harmful instances is not new and well known and applied in the recent times. Hence I feel the technical novelty is not solid and limited in some aspects. However saying that, applications of existing influence functions are not straightforward and hence that’s one point to be noted.


In general, the paper is well-organized, a good study on influence based relabeling and has sufficient empirical studies to backup it’s proposed formulation.

**Summary Of The Paper:**

The paper uses influence functions from robust statistics to perform two tasks in order to mitigate label noise and generalize better: (i) Identify harmful instances via IF; (ii) Relabel the harmful instances to have better generalization.


**Summary Of The Review:**

The paper does an end to end study of using influence functions for relabeling instances to achieve a lower test-loss. The technical novelty is slightly limited as the formulations are extensions of current influence functions, however the paper is supported by strong empirical studies (including limitations about RDIA in deep models).

---

> ### Author Response · Authors · 2021-11-12
> **Response to Reviewer 136B**
>
> Thank you for your valuable feedback, and for appreciating the motivation of our work and the technical contribution. We address your concerns below.
>
>
>
> >The technique of using influence function for identifying harmful instances is not new and well known and applied in the recent times. Hence I feel the technical novelty is not solid and limited in some aspects. However saying that, applications of existing influence functions are not straightforward and hence that’s one point to be noted.
>
>
>
> We would like to clarify the novelty of our approach in the following aspects:
>
> - **Labeling perturbation.** Previous influence-based approaches mainly focus on perturbing the loss term of the harmful samples (i.e. downweight all the harmful samples). To the best of our knowledge, we are the first to perform fine-grained perturbation on the harmful training sample’s label and evaluate its influence on model’s performance using the influence function.
>
> - **Theoretical guarantee.** Developing an influence-based relabeling function that theoretically guarantees to improve model’s performance and is applicable for any classification tasks is a non-trivial task. We also empirically show that our relabeling approach could achieve lower test loss than various existing influence-based resampling approaches. We believe that our work can inspire new research ideas to explore novel relabeling approaches for solving training bias.
>
> - **Relabeling on Deep Models.** As pointed out by Reviewer UYUF, the cost of estimating influence functions on non-linear models is a well-known challenge. We extend RDIA to RDIA-LS that combines the small-loss trick with relabeling and works well on deep models. Table 6,7,8,9 show that RDIA-LS is effective and efficient to handle training data with corrupted labels for deep learning models, compared with the baseline which are proposed to specifically solve the noisy labels for deep model. In this way, we believe that our work can inspire new research ideas to derive the extension of other influence-based approaches.
>
>
>
> We would like to thank the reviewer again for his/her valuable comments, which would improve the quality of our paper.
>
>
>
> We hope we have addressed your questions; please let us know if there are any additional concerns.

---

### Official Review · Reviewer_UYUF · 2021-11-03

**Correctness:** 4
**Technical Novelty And Significance:** 3
**Empirical Novelty And Significance:** 3
**Recommendation:** 8
**Confidence:** 4

**Main Review:**

The manuscript is well-written and clear. The proposed solution is
simple and clean with a sound theoretical justification.

The practical utility of the approach is unfortunately limited given
the cost of estimating influence functions on non-linear models. This
is a well-known problem, especially when applying influence function
to deep architectures. The authors address the problem by proposing to
replace influence functions with training loss while retaining their
relabeling scheme. I wonder how this simple strategy works in
comparison to their RDIA approach (the appendix does not report
comparisons in terms of accuracy), and under which conditions it could
fail (e.g. distribution shift rather than noise). Otherwise one has
the (probably false?) impression that all you need is training loss +
relabeling.

**Summary Of The Paper:**

The paper presents a relabeling scheme for binary and multiclass
classification tasks in which harmful training examples identified by
influence functions are relabeled to improve performance on a hold-out
set. The authors formally prove that the relabeling scheme determines
a reduction in loss wrt down-weighting or discarding examples, and
report experimental comparisons confirming the advantage of the
proposed strategy.

**Summary Of The Review:**

A simple and clean solution for influence-based data relabeling.

---

> ### Author Response · Authors · 2021-11-12
> **Response to Reviewer UYUF**
>
> Thank you for the positive review and encouraging feedback!
>
> In the appendix, we discussed the limitation of RDIA on deep models due to the erroneous influence estimations. As shown in the following table, we conduct the experiment on CIFAR10 dataset with a CNN (6 convolutional layers followed by 2 fully connected layers) and find the improvement of using RDIA to solve the label noise on deep model is limited. Hence, we extend RDIA to RDIA-LS by using training loss to identify noisy samples. We find that RDIA-LS could achieve better prediction performance than RDIA for combating the noisy labels on deep models.
>
> |        CIFAR10       | ERM   | RDIA  | RDIA-LS |
> | ----------------- | ----- | ----- | ------- |
> | 20% noise | 71.84 | 74.26 | 82.94   |
> | 40% noise | 55.62 | 58.41 | 77.26   |
>
>
>
> Following your suggestion, we have discussed the conditions that RDIA-LS could fail and also revised the paper to add this discussion in the last paragraph of Appendix G. We would like to clarify that RDIA-LS relies on the small-loss trick (only holds for deep models) that the samples with larger training loss may contain the corrupted labels. In this way, RDIA-LS is only suitable for deep models against corrupted labels. RDIA-LS would fail in distribution shift since the small-loss trick does not hold any more. However, we believe that if we could find some reliable criterion to select the harmful samples for the specific problem (For example, [1],[2] find that the samples with smaller training loss may hurt the model performance for class imbalance problem), our approach could still relabel these biased samples towards better model performance. Thanks for your valuable comment, we will continue research in this direction in the future.
>
> Reference
> [1] Learning to reweight examples for robust deep learning. ICML 2018
> [2] Meta-Weight-Net: Learning an Explicit Mapping for Sample Weighting. NIPS 2019

---

### Official Review · Reviewer_XS8o · 2021-11-03

**Correctness:** 4
**Technical Novelty And Significance:** 2
**Empirical Novelty And Significance:** 2
**Recommendation:** 8
**Confidence:** 3

**Details Of Ethics Concerns:**

Not sure if data resampling can have a negative impact and be a fairness concern if we are dealing with imbalanced datasets where the datasets represent people where some minorities are less represented in such datasets.
I have posted this as part of a possible discussion with the authors, to see what they think.



**Main Review:**

I believe the paper is very well-written and structured. I appreciate that the authors had taken time and consideration into writing an abstract and introduction that clearly motivates the problem in hand, gives enough background into the problem, and that clearly explains the solutions and the experimental results.

The paper seems to be solidly based in theoretical proofs of their methods, together with experimental results comparing it to some baselines plus the state-of-the-art approach. I liked that the limitations are clearly explained in the appendix.
All in all I think is a good paper.

Regarding weaknesses of the paper that I believe could help at improve the paper if addressed.
- First, is not until the reader reaches section three that the reader realizes that the authors use influence analysis on the validation set rather than on the test set. I think that the authors should improve the consistency across the paper of using the term "test set". Throughout the introduction, abstract, background, and part of the methodology, the validation set is referred as the test set. Which then is clarified to be the validation set, since the test set is only used for a proper test evaluation (without altering the training set). I would suggest to either clarify that at the beginning, or simply use validation set instead of test set.
- Algorithm for RDIA such as done for RDIA-LS (Algorithm 1) in the Appendix E. I believe this will help portray more details for further reproducibility.
- Regarding reproducibility, the authors mention that the code is available in the Appendix. Do the authors mean the programming code? If so, I couldn't find a link there. A link with the source code would be beneficial for the same reasons as the point above.
- My last and probably the most relevant concern is the following. Imagine we have an imbalanced dataset, and that the algorithm is not able to classify well that portion of the dataset with fewer examples. The problem in this case is that we don't have enough data of some particular group of instances, needing some sort of solution like data augmentation (for example).
However, if we would apply the authors solution (or the other authors solution for that matter) we would be treating these examples as incorrect. In the authors case the positive aspect is that the examples won't be removed (as with some of the others solutions). But, how would relabeling those examples help the model learn about this specific category of instances? I am thinking on the lines of a dataset where we have sensitive attributes e.g race, gender. Where some of the people are less represented. From my perspective this solution might have a negative impact in the generalization aspect since the model might not really learn those people that might look different from the rest. Or it could be the case where it helps at mitigating this bias.
This comment is more of an opening for a discussion with the authors, rather than "something to fix" . It would be great if the authors could just comment what they think on this matter during the discussion period. Since I am sure they have thought about this aspect, and I wouldn't want to have missed or misinterpreted some part of the paper.
Just to clarify, my scoring hasn't been influenced by this last comment.

**Summary Of The Paper:**

The authors present an approach and framework to mitigate training biases by combining influence functions and data relabeling.
The idea behind training biases is that part of the data that is used to train the model does not accurately represent the real data distribution seen in the test set. Thus, having a mismatch between training and test data. This creates a generalization problem for the machine learning model.

Other authors have used different resampling approaches to try and address this problem, relying on the training loss and then relabeling the data; or by using influence functions and changing the weight of the harmful examples so that the effect on the test loss is lower.
The current authors combine both approaches, and present a framework that relabels harmful training data based on influence functions (on the test set).

The results of their experiments show that they are able to reduce the test loss compared to other data resampling approaches.

**Summary Of The Review:**

I think this is a good paper with merits to be included as part of the conference proceedings.
I believe that the paper is well-written, the claims are well-supported, and the experiments seem correct.


I believe a few things can be improved (as mentioned above)
The reason I don't give a higher score is because I believe that the methodology does not seem to be too novel and the results are marginally better, except for one or two datasets.

---

> ### Author Response · Authors · 2021-11-12
> **Response to Reviewer XS8o**
>
> Thank you for the positive review and encouraging feedback! We reply individually to each raised point below
>
>
>
> > I would suggest to either clarify that at the beginning, or simply use validation set instead of test set.
>
>
>
> Thank you for pointing out this issue. We have revised the penultimate paragraph of the Introduction to clarify that we actually use influence analysis on the validation set rather than on the test set.
>
>
>
> > Algorithm for RDIA such as done for RDIA-LS (Algorithm 1) in the Appendix E. I believe this will help portray more details for further reproducibility.
>
>
>
> Thank you for your suggestion. We have added the algorithm for RDIA in Appendix A and kindly referred to that in the first paragraph of methodology.
>
>
>
> > Regarding reproducibility, a link with the source code would be beneficial for the same reasons as the point above.
>
>
>
> Sorry for the confusion. We uploaded the code repository in the Supplementary Material for reproducibility. We promise to include the URL link to our source code in the paper upon acceptance.
>
>
>
> > Imagine we have an imbalanced dataset, and that the algorithm is not able to classify well that portion of the dataset with fewer examples. The problem in this case is that we don't have enough data of some particular group of instances, needing some sort of solution like data augmentation (for example).
>
>
>
> Very inspiring comment and we are happy to discuss the potential benefits of applying our approach on imbalanced dataset. In fact, class imbalance is a typical distribution mismatch problem if the test set has balanced classes ($P_{train}$  is different from  $P_{test}$). According to [1] and [2], influence function is effective to identify the training examples that are responsible for the distribution mismatch. Existing work [3] thus has used influence function to solve the problem of class imbalance. It is important to notice that, if we use influence function to identify harmful samples over imbalanced training set, **most of the identified harmful samples lie in the majority class** which is in line with the analyze in [1] and the experiment results in [3]. In this way, influence-based reweighting approach (e.g. [3]) could improve the model’s predictive performance to the minority class by down-weighting the samples in the majority class. Since we have proved that our approach could achieve lower test loss than the reweighting approach, we think our approach could also solve the class imbalance problem to a certain extent. We provide some intuitive explanations here. Our influence-based relabeling approach acts like adding a few “noises” in the majority class and tells the model to forget the specific pattern of the data corresponding to the majority class, which avoids overfitting to the majority class. This could also mitigate the overfitting of the decision boundary arising from the influential majority samples and thus improve the model generalization ability on the minority class. This idea is like some previous approaches [4],[5] that add noises or penalize the over-confident model predictions to avoid overfitting to the specific pattern of the data and then improve the generalization performance of the model.
>
>
>
> > The reason I don't give a higher score is because I believe that the methodology does not seem to be too novel and the results are marginally better, except for one or two datasets.
>
> Previous influence-based approaches mainly focus on perturbing the loss term of the harmful samples (i.e. downweight all the harmful samples). **To the best of our knowledge, we are the first to perform fine-grained perturbation on the harmful training sample’s label and evaluate its influence on model’s performance using the influence function.** Furthermore, developing an influence-based relabeling function that theoretically guarantees to improve model’s performance and is applicable for any classification tasks is a non-trivial task. We also empirically show that our relabeling approach could achieve lower test loss than various existing influence-based resampling approaches. **A promising outcome of our approach is that when the noise rates increase (most of training samples are harmful), our approach could outperform other baselines by a large margin (Figure 2).** We believe that our work can inspire new research ideas to explore novel relabeling approaches for solving training bias.
>
> We hope we have addressed your questions; please let us know if there are any additional concerns.
>
>
>
>  Reference
>
> [1] Understanding Black-box Predictions via Influence Functions. ICML 2017
>
> [2] Less Is Better: Unweighted Data Subsampling via Influence Function. AAAI 2020
>
> [3] Influence-Balanced Loss for Imbalanced Visual Classification. ICCV 2021
>
> [4] DisturbLabel: Regularizing CNN on the Loss Layer. CVPR 2016
>
> [5] Regularizing Neural Networks by Penalizing Confident Output Distributions. ICLR2017

---

> > ### Comment · Reviewer_XS8o · 2021-11-19
> > **Reply. Discussion**
> >
> > I would like to thank the authors for the answers to the comments and for the nice discussion.
> > What the authors comment make sense about imbalanced datasets, and how influence functions can help at mitigate the effects usually caused by the imbalance.
> > I particularly liked the intuitive explanations related to overfitting.
> >
> > Regarding the score and the novelty, I have decided to increase my score, since I might not have understood the full extent of their approach wrt its novelty and its comparison with the other two approaches that they built on.
> >
> > Good job.

---

### Decision · Program_Chairs · 2022-01-20

**Decision:**

Accept (Oral)

**Comment:**

All reviewers are very positive about this paper. The reviewer with the lowest score did independent experiments that show that the authors' method works well, and has had an extensive discussion with the authors that justifies a higher score. The paper is potentially very valuable to practitioners, since it shows how to compensate for a training set that is not representative of the test data.

Suggestion from the area chair to the authors: Briefly discuss the relationship between influence scores and propensity scores, which are standard in the literature on causal modeling and on sample selection bias, as in https://jmlr.csail.mit.edu/papers/volume10/bickel09a/bickel09a.pdf for example.